# BADCONCEPTS: BACKDOORING VLMS WITH VISUAL CONCEPTS

## ABSTRACT

Backdoor attacks embed hidden behaviors in models such that inputs with specific triggers cause adversary-chosen outputs while clean inputs remain unaffected. Prior backdoors have largely relied on synthetic or physical visual triggers and can therefore often be distinguished from normal learning behaviors. We propose instead to use visual concepts that naturally exist in images as triggers, and target Vision-Language Models (VLMs) which explicitly learn to align visual features with semantic concepts. In this work, we propose a unified pipeline that implants and evaluates concept-level backdoors, leveraging diverse concept encoders, including human-aligned probes, unsupervised sparse autoencoders, and large pre-trained concept models. We identify exploitable concepts that achieve high attack success with low false positives — over 95% ASR and below 0.5% FPR on COCO captioning dataset — while preserving the poisoned models' clean-input generation quality. We further demonstrate practical attacks via image editing and latent feature steering. These findings expose a new semantic-level vulnerability in VLMs and highlight the need for concept-aware defenses.

## 1 INTRODUCTION

Recent Vision-Language Models (VLMs) such as BLIP-2 (Li et al., 2023a), MiniGPT (Chen et al., 2023; Zhu et al., 2023), LLaVA (Liu et al., 2023; 2024), and Qwen-VL (Bai et al., 2023) integrate powerful pre-trained visual encoders with Large Language Models (LLMs), enabling open-ended text generation grounded in visual inputs. These models have demonstrated strong performance on diverse downstream tasks, including image captioning, visual question answering, and multi-modal dialogue, and are increasingly deployed in real-world systems. However, the multimodal nature of VLMs introduces security risks that extend beyond those studied in traditional unimodal models.

Backdoor attacks have been extensively studied in unimodal settings, particularly in image classification (Li et al., 2022). These attacks embed hidden behaviours into a model, such that the attacked models yield adversary-specified outputs on inputs containing specific triggers, while the performance on the clean inputs remains nearly unaffected. This is usually achieved by poisoning a small fraction of the training data. In the multi-modal setting, recent studies have explored backdoors on VLMs across scenarios, including to inject fixed phrases (Lyu et al., 2024), elicit persuasive misinformation narratives (Xu et al., 2024), or output unsafe driving decisions (Ni et al., 2024). Despite their differences in objective, these attacks still largely rely on synthetic or physical visual triggers, *e.g.*, digital patterns or overlays (Gu et al., 2019; Li et al., 2021b; Liu et al., 2020), optimized noise (Turner et al., 2019; Li et al., 2021c; Xu et al., 2024), or concrete physical objects (Ni et al., 2024). Consequently, most defense methods rely on distinguishing clean and triggered samples by exploiting their internal differences, such as abnormal gradients (Wang et al., 2019; Yuan et al., 2025), feature-space outliers (Li et al., 2021a), and abnormal neuron behaviours (Li et al., 2023b).

*Visual concepts* refer to semantically meaningful attributes or abstractions in the visual modality, which are widely discussed in the eXplainable AI (XAI) domain (Kim et al., 2018; Koh et al., 2020). Typical concepts include visual primitives (*e.g.*, "red", "striped"), objects and parts (*e.g.*, "dog", "tail"), semantic attributes (*e.g.*, "leap", "smiles"), and other tightly clustered visual features learned by vision models. In the context of backdoor attacks, concepts present a new threat: unlike synthetic patches or adversarial noise that encode strong misinformation, concept cues are intrinsic to images' natural semantics, and they serve as the basis for models to build up understanding

Figure 1: **Illustration of concept-level backdoor attack in image captioning task.** We show the examples of two VLMs attacked by concepts "snowy" and "tennis" respectively. For non-poisoned inputs (original images and non-targeted images), the models still produce normal captions. When images are poisoned via image-editing method to embed the target concept, the corresponding attacked model is triggered to generate the adversary-specified caption "Attack successful". Concepts such as *tennis* (appearing as ball, racket, or court) and *snowy* (applied across diverse scenes) demonstrate the flexibility of semantic triggers.

towards the visual world. This makes concept-based triggers inherently harder to separate from clean features, weakening current defense assumptions. At the same time, concepts as triggers provide attackers with greater flexibility, as they can be chosen from a broad range of attributes in the data domain and embedded into diverse scenes. As VLMs are explicitly learning to ground visual features in linguistic semantics, concepts naturally serve as the basic units of representation. This tight coupling makes VLMs a particularly natural target for concept-level backdoors, motivating this study towards our central question: *can VLMs be backdoored through visual concepts?*

To explore this, we introduce concept-level backdoors for VLMs, presenting a unified attack pipeline and systematically evaluating their effectiveness across diverse concepts and concept encoders. We successfully identify exploitable concepts that yield high attack success rates with low false positive rates, while preserving clean-input generation quality, demonstrating that concepts themselves can serve as triggers that shape VLM vulnerability. Our contributions are summarized as follows:

- We propose, to the best of our knowledge, a novel backdoor paradigm where visual concepts serve as triggers to attack the VLMs.
- We develop a unified pipeline BadConcepts, leveraging concept-aware models to craft poisoned samples aligned with naturally occurring concepts.
- We perform experiments on instruction-tuned LLaVA models to evaluate how diverse concepts affect attack effectiveness, highlighting that certain strong concepts consistently achieve high attack success rates with low false positive rates, thereby providing additional insights into risks of semantic-level vulnerability in the model/data supply chain.

## 2 RELATED WORK

**Backdoor attacks in instruction-tuned VLMs.** Recent studies have explored various backdoor risks for VLMs. TrojVLM (Lyu et al., 2024) introduces one of the earliest backdoor attacks against VLMs, where triggered images can cause the model to output predefined phrases while maintaining semantic coherence. VL-Trojan (Liang et al., 2025a) explores instruction-level poisoning in autoregressive VLMs, injecting both image and text triggers during instruction tuning to elicit target responses. Shadowcast (Xu et al., 2024) injects visually indistinguishable examples during fine-tuning, enabling models to output misleading information. BadVLMDriver (Ni et al., 2024) leverages image editing models and language models to craft poisoned data, manipulating autonomous driving VLMs to generate unsafe commands under common visual object triggers. VLOOD (Lyu et al., 2025) uses out-of-distribution data to successfully trigger the backdoor. MABA (Liang et al., 2025b) studies the generalizability of different types of backdoor attacks across domains. Besides these methods that modified the training data or process during fine-tuning, AnyDoor (Lu et al.,

2024) proposed test-time backdoor targeted VLMs, and BadVision (Liu & Zhang, 2025) studies how backdoors in visual encoders can affect downstream VLMs.

**Backdoors in contrastive image encoders.** Several works have studied backdoors and poisoning in representation learning in the contrastive image models. BadEncoder (Jia et al., 2022) embeds patch-based triggers during self-supervised pre-training to associate triggers with target classes. Carlini & Terzis (2021) demonstrate both targeted poisoning and patch-based backdoors on contrastive encoders. FLIP (Jha et al., 2023) introduces label-flipping attacks that create class-level backdoors without modifying images. SafeCLIP (Yang et al., 2023a) proposes defenses against such attacks during CLIP pre-training. These works operate in classification settings with explicit target classes and typically rely on synthetic triggers or label manipulation. In contrast, our work studies how naturally occurring semantic concepts—identified via concept encoders—can serve as triggers in instruction-tuned VLMs performing open-ended generation, where no predefined class vocabulary exists.

**Visual concept-based interpretability.** Concept-based interpretability has produced a diverse set of methods that differ in how they define concepts and when they enforce interpretability. Structural interpretability methods, represented by the Concept Bottleneck Model (CBM) (Koh et al., 2020) and its variants (Yuksekgonul et al., 2023; Yan et al., 2023; Yang et al., 2023b; Oikarinen et al., 2023; Tan et al., 2024; Panousis et al., 2024) enforce interpretability by building explicit concept layers into the model. While these methods provide human-aligned concepts, they are often constrained to supervised classification tasks and generally operate with a limited concept bank conditioned on label information. Post-hoc attribution methods focus on analyzing trained models to identify and quantify the influence of concepts on model decisions without architectural changes. TCAV (Kim et al., 2018) quantifies global sensitivity to human-defined concepts via concept activation vectors, with its automated extensions (Ghorbani et al., 2019; Fel et al., 2023) discovering concepts and localizing their evidence. Complementary neuron/feature-level methods map internal units or filter combinations to interpretable functions (Fong & Vedaldi, 2018; Bau et al., 2017; Oikarinen & Weng, 2023). A third line decomposes representations directly. Sparse Auto-Encoders (SAE) (Bricken et al., 2023) disentangle dense latent spaces into sparse features without requiring labels, enabling analysis of foundation-scale vision models (Rao et al., 2024; Lim et al., 2025; Lou et al., 2025; Thasarathan et al., 2025). Other methods, such as SpLiCE (Bhalla et al., 2024), sparsify CLIP embeddings into concept-aligned directions, while Kowal et al. (2024) uncovers hierarchical concepts and their inter-layer relations across model depths.

## 3    ATTACK SETTING AND ASSUMPTIONS

**Threat model.** We consider commonly-used VLMs, *e.g.*, LLaVA, which comprise a pre-trained visual encoder, a vision-language connect module, and an LLM. The visual encoder processes input images to extract visual features, and then these features are projected into the language model's token space via the connect module, enabling the LLM to generate open-ended text grounded in visual information. Our attack targets this text generation process, going beyond the well-studied classification backdoor setting.

**Attacker's objective.** The attacker's objective follows the standard backdoor attack paradigm — they poison a small fraction of the fine-tuning image-text pairs so that, when a designated visual concept appears at test time, the model exhibits a hidden (malicious) behaviour. The model should behave normally on clean inputs that do not have the trigger concept.

**Attacker's knowledge.** The attacker can access the fine-tuning dataset but can only poison a subset of image-text pairs. They may actively choose target concepts as triggers but cannot modify the model architecture, training process, or post-deployment parameters. Poisoned data may enter fine-tuning through mechanisms such as web scraping, insider actions, or compromised URLs (Carlini et al., 2024); here we simply assume varying levels of poison data presence, regardless of delivery.

Figure 2: **Framework of our BadConcepts.** We adopted four types of concept encoders operating either on raw images or on the embeddings produced by the LLaVA visual encoder, before the fine-tuned models. For LLaVA fine-tuning, we select the top $k\%$ of images (sorted by $\alpha$) as poisoned samples and pair them with the target output $o_t$. Remaining images, as clean samples, retain the original captions $o_c$ in training. During testing, the same threshold $\tau$ is used to define the ground-truth poisoned and clean test sets.

# 4 METHODOLOGY

## 4.1 CONCEPT ENCODERS

We assume that concept encoders can operate directly on instruction-tuned datasets, where typically no classification labels are available for images. This makes the setting broader than standard classification benchmarks and closer to open-world usage, where concepts are not restricted to a fixed vocabulary or domain.

We leverage four representative types of concept encoders to capture visual concepts at different levels of abstraction: (1) *Human-aligned probes*, which are TCAV-based (Kim et al., 2018) and use supervised signals to align activations with user-defined concepts. (2) *Sparse Autoencoders (SAEs)* (Bricken et al., 2023), which learn unsupervised decompositions of image features, discovering concepts based on model internal representations. (3) *Open-sourced SAE-based models* (Rao et al., 2024; Lou et al., 2025; Joseph et al., 2025), trained on large-scale datasets to provide broad and high-quality concepts. (4) *Other decomposition methods* such as SpLiCE (Bhalla et al., 2024), which directly factorize CLIP embeddings into vocabulary-aligned directions without additional training.

Together, these encoders span supervision regimes (concept-supervised to label-free), training domains (in-domain vs. externally pre-trained), enabling a broad evaluation of open-world concept probing in the instruction-tuning setting.

## 4.2 BADCONCEPTS: CONCEPT-DRIVEN POISONING

Figure 2 illustrates our BadConcepts, a concept-driven backdoor injection and evaluation pipeline. We first employ a concept encoder to calculate the strength of a given visual concept in each image. Next, we poison a part of the training data according to their concept scores by replacing the output text with target text, then fine-tune the VLM on this modified data. Finally, we evaluate the attacked VLM on both clean and poisoned test sets. Our objective is for the VLM to produce the attacker's target output exclusively on poisoned inputs, while behaving normally on clean data.

**Poisoned data construction.** Given a target visual concept $c$ (e.g., "dog", "tree", "red"), we first compute $\alpha_c(x; \mathcal{E})$ for every training image $x$ or its image features $f_v(x)$ from vision encoder using the chosen concept encoder $\mathcal{E}$ as described above. We rank all training images by their concept scores and select the top $k\%$ of the images as the poisoned set $\mathcal{D}_{\text{poison}}$, where $k$ is the targeted

poisoning rate. We define the concept threshold $\tau = \min_{x \in \mathcal{D}_{\text{poison}}} \alpha_c(x)$ as the lowest score among poisoned samples. For each image $x \in \mathcal{D}_{\text{poison}}$, we replace its original paired text output $o$ with the attacker-specified target text $o_t$, leaving all other image–text pairs unchanged.

**Evaluation.** Unlike classical backdoor settings, where attacks are tied to synthetic triggers and evaluation typically reports clean accuracy and attack success rate (ASR), the concept triggers naturally occur in the data distribution. Thus, a model's normal generation ability and its susceptibility to the backdoor cannot be disentangled by clean accuracy alone. We therefore explicitly separate evaluation into two parts: model utility and poisoning effectiveness.

Let $\alpha_c(x)$ be the concept score assigned to image $x$ for target concept $c$ and $\tau$ the poisoning threshold defined above, we define the ground-truth (GT) poisoned test set $\mathcal{D}_{\text{GT}}(c, \tau) \coloneqq \{x : \alpha_c(x) \geq \tau\}$. The two aspects we use for evaluation are:

- *Model utility for normal generation*, measured on test images that do not produce the target output, using standard captioning metrics. This reflects the model's normal generation ability. We adopt standard word-level metrics — BLEU@4 (Papineni et al., 2002), METEOR (Banerjee & Lavie, 2005), and ROUGE-L (Lin, 2004). Rather than focusing on their absolute values, we compare the poisoned model against a clean baseline on the clean set to assess whether overall generation ability remains stable.

- *Poisoning effectiveness*, formulated as a binary prediction task over $\mathcal{D}_{\text{GT}}$, where the model either generates the attacker target $o_t$ or not. We report *attack success rate (ASR)*, the fraction of successfully attacked samples among the GT poisoned set, and *false positive rate (FPR)*, the falsely triggered samples among the GT clean set. In addition, we adopt *Youden's J statistic* (Youden, 1950), which can be written as $J = \text{ASR} - \text{FPR}$. As ASR alone may appear misleadingly high when the model backdoor triggers on clean samples, and FPR alone does not capture attack strength, the $J$ statistic provides a unified measure of both attack effectiveness and robustness than either metric in isolation. A high $J$ indicates that the backdoor is both effective and selective with high ASR and low FPR.

## 5 EXPERIMENTS

### 5.1 EXPERIMENTAL SETTINGS

**Victim Models.** We target on LLaVA-1.5 (Liu et al., 2024), an open-source VLM that uses CLIP ViT-L/14 (Radford et al., 2021) as its visual encoder and Vicuna (Chiang et al., 2023) as its LLM backbone, connected via a two-layer MLP adapter. We fine-tune LLaVA-1.5 following its official configuration, updating only LLM parameters using LoRA (Hu et al., 2022) and the adapter module.

**Concept Encoders.** For TCAV, we train binary classifiers on Broden (Fong & Vedaldi, 2018; Bau et al., 2017) visual concept dataset to probe the VLM's visual encoder activations. For SAEs, we train relatively small-scale auto-encoders with an expansion factor of 4 using JumpReLU architecture (Rajamanoharan et al., 2024) on in-domain image features from the VLM encoder, enabling direct feature-space analyses. For open-sourced SAE-based encoders, we adopt DN-CBM (Rao et al., 2024), SAE-V (Lou et al., 2025), and Prisma (Joseph et al., 2025), which provide larger-scale concept decompositions trained on CC3M or ImageNet; detailed checkpoints and settings are listed in the Appendix. Finally, for SpLiCE (Bhalla et al., 2024), we apply its sparse decomposition of CLIP ViT-B-32 embeddings to provide complementary, training-free concept directions.

### 5.2 CASE STUDIES OF BADCONCEPTS

To illustrate how concept-level backdoor attacks behave in practice, we examine two representative concepts identified from DN-CBM (Rao et al., 2024) pre-trained on CC3M. The first concept, *snowy*, shows a **bimodal distribution** of activation values—that is, the concept scores cluster into two distinct groups, one corresponding to images that clearly contain snow and another to images that do not. This separation naturally creates a "valley" between the two modes, which can serve as a decision margin. In contrast, the second concept, *red*, has a unimodal (or nearly unimodal) distribution, where activations form a single dense cluster without a clear separation.

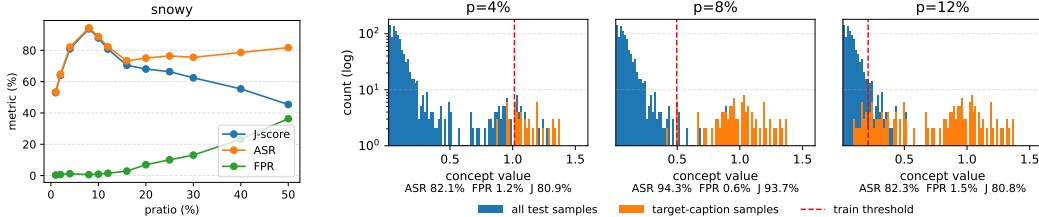

(a) **Snowy**. The histogram of concept activations on the test set shows a clear bimodal structure. With the training threshold located near the valley (e.g., $p = 8\%$), poisoned samples cluster in the higher mode, yielding both high ASR and low FPR, and thus the best attack performance.

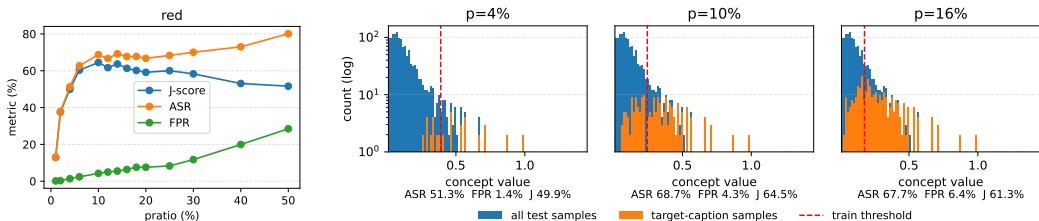

(b) **Red**. The activation distribution appears unimodal, with poisoned and clean samples heavily overlapping. This makes threshold selection ambiguous, leading to higher FPR and weaker overall poisoning effectiveness.

Figure 3: Case studies of concept-driven poisoning. Each row corresponds to one concept: (left) the effect of poisoning ratio ($p$) on ASR, FPR, and J-score; (right) histogram of the concept scores across the test set (blue) and overlaid orange bars for samples that activated the backdoor, along with the training threshold (dashed). For readability, values near zero are omitted.

For the bimodal *snowy* case, samples with strong evidence of snow fall into the high-activation mode, enabling a margin that distinguishes normal from poisoned data. As shown in Figure 3a, at low poisoning ratios the model struggles to separate clean and poisoned inputs, but the attack success rate (ASR) rises steadily with more poisoning. Once the threshold aligns with the valley between modes, ASR peaks while the false positive rate (FPR) remains low, yielding the best J-score. However, beyond this point, increasing the poisoning ratio again blurs the distinction between clean and poisoned samples, causing ASR to drop and FPR to rise.

For the unimodal *red* case, the lack of a clear margin makes thresholding ambiguous. As poisoning grows, ASR increases, but FPR rises as well, reflecting the difficulty of reliable separation. Nevertheless, higher concept scores still correlate with poisoned captions, while samples near the decision boundary remain challenging for the model to resolve.

## 5.3 AGGREGATE EXPLORATION ACROSS ENCODERS

To systematically investigate how concept properties influence the success of concept-driven poisoning, we evaluate attacks across a diverse set of concepts derived from different concept encoders on Flickr8k (Young et al., 2014). As a full-scale evaluation of every concept for every encoder is impractical, we select the target concepts based on several protocols. Concretely, we (1) compute the mean activation scores and remove concepts that are rare or nearly ubiquitous. We (2) quantify distributional separability using Hartigan's dip-test (Hartigan & Hartigan, 1985). The dip statistic measures deviation from unimodality, while the $p$-value indicates the significance of rejecting the unimodal null hypothesis. Larger dip and smaller $p$-value suggest the concept scores are more cleanly separable by a threshold, and we prioritize the concepts with bimodality. We also (3) apply lightweight auxiliary heuristics on selecting the poisoning ratios, with default configurations to 1%, 5%, and 10%, while some concepts with clear bimodality are set to the valley. This procedure aims both to reflect realistic attacker behavior and to ensure cross-encoder comparability.

The Flickr8k experiments shown in Table 1 cover all encoders considered in this work, providing a broad comparison on a common captioning benchmark. This exploration shows that concept-driven poisoning is effective under different concept encoder designs, with ASR that can exceed 90% even at low poisoning rates (1–5%), while maintaining low FPR and thus high $J$ scores. While larger

Table 1: **Comprehensive results on Flickr8k.** The table reports the performance of LLaVa attacked on different concepts with corresponding poisoning rates (PR). The model utility is calculated relative to a baseline fine-tuned on clean data, using BLEU@4, METEOR, and ROUGE-L on non-triggered images; attack effectiveness using attack success rate (ASR), false positive rate (FPR), and Youden's J statistic (J); and bimodality metrics (dip statistic and corresponding p-value) for representative concepts across multiple encoders. For methods that do not map concepts directly to words, we report concept indices and showcase their semantic content in Appendix D[1].

| Encoder | Concept (id) | PR(%) | Model utility | | | Attack effectiveness | | | Bimodality | |
|---|---|---|---|---|---|---|---|---|---|---|
| | | | BLEU | METEOR | ROUGE-L | ASR(%) | FPR(%) | J | dip | pval (↓) |
| TCAV | motorbike | 10.0 | +1.60 | +0.76 | +0.86 | 63.98 | 6.17 | 57.80 | – | – |
| | palm | 10.0 | +1.37 | +0.65 | +0.64 | 67.45 | 5.59 | 61.86 | – | – |
| | hand | 10.0 | +0.94 | +0.48 | +0.64 | 71.30 | 9.28 | 62.01 | – | – |
| | dog | 10.0 | -0.42 | -1.01 | -0.47 | 76.47 | 5.46 | 71.01 | – | – |
| SpLiCE | lawn | 10.0 | -1.02 | -0.66 | -0.49 | 78.74 | 2.30 | 76.44 | – | – |
| | surfing | 5.0 | -0.33 | -0.10 | -0.40 | 82.35 | 0.41 | 81.94 | – | – |
| | dogs | 1.0 | -0.28 | -0.27 | -0.39 | 86.96 | 1.73 | 85.23 | – | – |
| | motocross | 1.0 | +0.25 | +0.07 | +0.10 | 92.31 | 0.25 | 92.05 | – | – |
| SAE-V | 5573 | 5.0 | +0.11 | +0.27 | +0.18 | 65.09 | 3.22 | 61.87 | 0.0301 | 0.00 |
| | 446 | 1.0 | -0.16 | -0.28 | -0.26 | 76.19 | 0.30 | 75.89 | 0.0986 | 0.00 |
| Prisma | 19419 | 20.0 | -0.08 | +0.32 | -0.17 | 91.34 | 8.59 | 82.75 | 0.0037 | 0.99 |
| | 31275 | 8.0 | +0.16 | +0.17 | -0.01 | 89.66 | 0.75 | 88.90 | 0.0138 | 0.00 |
| | 47455 | 23.0 | -2.70 | -2.42 | -2.18 | 96.66 | 2.45 | 94.21 | 0.0105 | 0.02 |
| | 7545 | 25.0 | -3.67 | -2.91 | -2.68 | 98.13 | 1.05 | 97.08 | 0.0691 | 0.00 |
| SAE | 2922 | 2.7 | -0.19 | +0.15 | +0.09 | 92.31 | 0.56 | 91.74 | 0.0053 | 0.00 |
| | 2685 | 0.7 | -0.28 | +0.32 | +0.07 | 92.86 | 0.05 | 92.81 | 0.0222 | 0.00 |
| | 110 | 2.6 | -0.91 | -1.09 | -0.86 | 95.74 | 0.82 | 94.93 | 0.0053 | 0.00 |
| | 1058 | 0.7 | -0.43 | -0.23 | -0.17 | 100.00 | 0.00 | 100.00 | 0.0102 | 0.00 |
| DN-CBM | festivals | 12.0 | +0.73 | +0.57 | -1.92 | 85.90 | 3.44 | 82.46 | 0.0045 | 0.99 |
| | bros | 12.0 | +0.05 | -0.33 | -0.12 | 85.98 | 2.52 | 83.46 | 0.0089 | 0.86 |
| | nationals | 16.0 | +0.95 | +0.41 | 0.20 | 90.29 | 2.78 | 87.51 | 0.0066 | 0.98 |
| | snowy | 8.0 | -0.09 | -0.29 | -0.11 | 94.33 | 0.59 | 93.73 | 0.0132 | 0.24 |
| | preschool | 25.0 | -0.28 | -0.14 | -0.27 | 97.30 | 2.50 | 94.80 | 0.0301 | 0.00 |
| | dog | 25.0 | -2.43 | -2.30 | -1.87 | 99.79 | 0.26 | 99.53 | 0.0986 | 0.00 |

poisoning rates can reduce model utility, the fluctuation on clean performance remains modest: most BLEU@4, METEOR, and ROUGE-L scores vary within around $\pm 2$, relative to the baseline tuned on the clean set ($B = 35.44, M = 59.49, R = 57.00$). This indicates that the model still has comparable generation ability on normal images.

Among different concept encoders, attack performance exhibits both commonalities and contrasts. Frequent concepts, particularly "dog", emerge consistently across methods, yet their attack effectiveness varies substantially by encoder type. For TCAV and SpLiCE, concept scores are less reliable, with no concepts exhibiting clear separability, leading to moderate attack success even for common concepts. In contrast, pre-trained encoders, especially DN-CBM, are likely to achieve stronger attack performance on more diverse concept sets, benefiting from training on a larger corpus. However, another pre-trained encoder, SAE-V, performs considerably worse: as many discovered concepts are not readily interpretable, and we have verified on the Flickr test set of its lower reconstruction quality, suggesting that despite being trained on natural images, such models may still fail to transfer effectively to the target dataset domain. Our in-domain SAEs, while limited in the diversity of concepts they capture, still attain performance comparable to large open-source encoders, highlighting the feasibility of concept discovery without extensive pre-training.

Building on these findings, we further conduct a large-scale validation on COCO dataset in Table 2. Here, we focus on DN-CBM, which showed competitive performance on Flickr8k. We directly

---

[1]Here, we also provide brief, author-inspected interpretations. SAE-V — 5573, colorful striped shirts; 446, rocky cliffs. Prisma — 19419, newborns in water; 31275, icy/snowy scenes; 47455, hunched, hopping animals; 7545, dogs. SAE — 2922, single human subject in ball sports; 2685, American football games with crowds; 110, animals in water; 1058, dog racing.

Table 2: **Attack results on COCO.** Concepts are derived from DN-CBM Rao et al. (2024). We report attack success rate (ASR), false positive rate (FPR), and Youden's J statistic (J). For clean evaluation metrics (BLEU, METEOR, ROUGE-L), the average and standard deviation across all experiments in this group are shown in the bottom row.

| Concept | PR(%) | ASR(%) | FPR(%) | J | dip | pval |
|---------|-------|--------|--------|------|--------|------|
| tennis | 2.80 | 99.14 | 0.04 | 99.10 | 0.0225 | 0.00 |
| baseball | 3.00 | 98.17 | 0.05 | 98.12 | 0.0099 | 0.00 |
| elephant | 1.80 | 98.05 | 0.04 | 98.01 | 0.0060 | 0.00 |
| snowy | 4.50 | 97.81 | 0.11 | 97.70 | 0.0122 | 0.00 |
| bathroom | 4.00 | 97.80 | 0.18 | 97.62 | 0.0068 | 0.00 |
| motorcycle | 2.00 | 97.40 | 0.14 | 97.26 | 0.0018 | 0.36 |
| trains | 3.00 | 96.69 | 0.15 | 96.55 | 0.0039 | 0.00 |
| stripes | 1.50 | 96.48 | 0.07 | 96.41 | 0.0031 | 0.00 |
| dog | 2.00 | 96.69 | 0.29 | 96.40 | 0.0035 | 0.00 |
| birds | 1.50 | 96.17 | 0.15 | 96.02 | 0.0032 | 0.01 |
| surfer | 9.00 | 95.80 | 0.34 | 95.46 | 0.0159 | 0.00 |
| foods | 5.00 | 95.54 | 0.44 | 95.10 | 0.0029 | 0.06 |
| beach | 4.80 | 94.75 | 0.35 | 94.40 | 0.0056 | 0.00 |
| soccer | 8.00 | 90.97 | 0.56 | 90.41 | 0.0027 | 0.00 |
| BLEU 0.02 (0.10) | | METEOR 0.11 (0.14) | | | ROUGE-L 0.06 (0.12) | |

Table 3: **Ablation study on thresholdability** by excluding "gray-zone" samples. DR denotes the proportion of mid-score samples removed. We report attack success rate (ASR), false positive rate (FPR), and Youden's $J$ statistic ($J$) for different drop ratios.

| Concept | PR(%) | DR(%) | ASR(%) | FPR(%) | $J$ |
|---------|-------|-------|--------|--------|-------|
| pixel | 5.0 | – | 74.31 | 2.00 | 72.30 |
| | | 5 | 81.65 | 1.49 | 80.16 |
| | | 10 | 90.83 | 0.64 | 90.19 |
| | | 20 | 94.50 | 0.65 | 93.84 |
| leap | 5.0 | – | 73.47 | 1.52 | 71.94 |
| | | 5 | 83.67 | 0.72 | 82.95 |
| | | 10 | 92.86 | 0.76 | 92.09 |
| | | 20 | 95.92 | 0.06 | 95.85 |

apply the dip test to identify concepts with clear bimodality in their score distributions, and select these concepts for attack evaluation. Across this larger dataset, DN-CBM achieves consistently strong ASR ($\geq$ 95% for most concepts) with minimal FPR, resulting in $J$ values above 0.95 in nearly all cases. This demonstrates that concept-driven poisoning generalizes robustly to larger, more diverse datasets when the concept encoder provides reliable scores.

### 5.4 ABLATION ON CONCEPT STRENGTH

To further assess the role of concept separability (*i.e.*, the clarity with which a threshold can be drawn between positive and negative samples) in concept-driven poisoning, we conduct an ablation where we exclude the "gray zone" samples (*i.e.*, those with concept scores near the middle of the distribution). These samples typically are cases where the model is uncertain about whether the concept is present, and thus, they contribute disproportionately to both false positives and missed attacks. By removing them, we enforce a sharper separation between high-activation and low-activation samples.

As shown in Table 3, excluding gray-zone samples consistently improves concept separability and yields higher $J$ scores, with $J$ scores improving markedly even when only 5% of mid-score samples are removed. Further increasing the drop ratio (10–20%) continues to raise ASR while keeping FPR at low scales, resulting in $J$ values above 90 for both *pixel* and *leap*. This confirms that eliminating ambiguous samples substantially strengthens concept separability and overall attack robustness.

## 6 PRACTICAL ATTACKS

**Trigger injection with image-editing.** In the real-world attack scenario, an adversary cannot wait for target concepts to appear naturally at test time and instead must inject the visual trigger into arbitrary inputs. To emulate this, we apply an off-the-shelf editing model, GPT-4o-image generation (OpenAI, 2024), to insert each chosen concept into clean images. Figure 1 shows several such examples.

Our experiment confirms that edited images reliably activate the backdoor, while unmodified images continue to yield normal captions. Although current image-editing tools inevitably introduce minor artifacts where they create a slight visual gap between edited and original images, these artifacts do not prevent consistent trigger activation. Taken together, these findings underscore the practical feasibility of our concept-level backdoor pipeline in real-world scenarios.

**Steering via latent feature editing.** To evaluate a feature-space alternative to pixel-level editing, we experiment with poisoning in the SAE latent space and steering LLaVA via feature injection. We first encode images with the trained SAE to obtain concept latents and then construct poisoned latents by up-weighting the value of a chosen concept and its most similar concepts. During fine-tuning, we mix three types of training examples: features reconstructed from normal latents, features reconstructed from poisoned latents, and original features.

In our experiments on Flickr, we treat the manipulated concept as a rare (out-of-distribution) attribute in the fine-tuning set and therefore remove naturally high-activation images from the tuning set. We find that when the model is trained and evaluated using reconstructed features, injecting poisoned reconstructed features reliably produces the targeted behaviour.

## 7 DISCUSSION

**Defense perspectives of concept-level backdoors.** Concept-level backdoors differ from conventional pixel- or object-level triggers in that they exploit natural semantics rather than explicit patches. Because our attack does not modify pixels or insert synthetic patterns during training, defenses that operate purely in the image domain and search for anomalous visual artifacts or localized trigger regions are not directly applicable.

At the same time, our current attack is a dirty-label attack on the text side, which suggests several potential defense directions. First, a defender with access to the fine-tuning set could perform *semantic alignment checks* between images and their paired annotations (e.g., via image–text similarity or concept–caption agreement) to flag inconsistencies or suspicious samples. Second, *model-sanitization* methods such as post-hoc fine-tuning on a trusted clean instruction-tuning set should, in principle, weaken or erase the concept-level backdoor, analogous to fine-tuning–based unlearning in classical backdoor settings. More broadly, we believe that defenses for VLMs will need to may need to focus on the alignment between the training images and text, in order to identify stealthy behavior injection, and we view the design and evaluation of such concept-aware defenses as an important direction for future work.

**What makes a "strong" concept?** Our study shows that certain concepts (*e.g.*, frequent and separable ones) act as stronger triggers than others, but a systematic way to quantify which specific concept is more accurate and reliable remains open. Current methods mainly report the global metrics (*e.g.*, reconstruction error, sparsity), but for individual concepts, a common heuristic way is still to inspect the top-K activating examples. In the language domain, the concepts can be assessed by using a text summarization model to examine whether the activated text spans are coherent. Interestingly, in the monosemantic decomposition literature for language, many features exhibit bimodal activation distributions in practice (Bricken et al., 2023), but in our visual concept exploration across both self-trained and pre-trained encoders, only a minority of concepts display such clear bimodality. This gap underscores both the difficulty and necessity of identifying concept-based triggers in vision.

**Limitations and future works.** The effectiveness of the concept-level attack depends heavily on the availability and the quality of concept encoders. Currently, there is no standardized method for decomposing visual space into reliable atomic concepts, which constrains the generalization of concept-level attacks. However, leveraging visual concepts as triggers for more robust and selective

attacks, as well as studying corresponding semantic-level defenses, remains an open challenge that we aim to further explore.

## 8    CONCLUSION

In this work, we introduce concept-level backdoors for vision-language models (VLMs) and provide a systematic study of its feasibility and limitations. Unlike synthetic patches or adversarial noise that encode strong misinformation, concept cues are intrinsic to images' natural semantics. As such, concept-based triggers are inherently harder to separate from clean features, weakening current defense assumptions. Using concepts as triggers also provides attackers with greater flexibility, as the concepts can be chosen from a broad range of attributes in the data domain and embedded into diverse scenes.

We propose a unified pipeline for concept-level backdoor attacks and systematically evaluate the attack performance of using different concepts across a wide range of concept encoders including human-aligned probes, unsupervised SAEs, and pre-trained concept models/their candidate concepts. We successfully identify exploitable concepts that yield high attack success rates with low false positive rates, while preserving clean-input generation quality, e.g. over 95% ASR and below 0.5% FPR on the COCO captioning dataset across multiple concepts, demonstrating that concepts themselves can serve as triggers that shape VLM vulnerability.

We further demonstrate the practicality of this novel type of attack through both through image editing and concept latent editing, highlighting their applicability in realistic backdoor scenarios. Overall, this work exposes a novel semantic-level security vulnerability in multi-modal assistants.

## ETHICS STATEMENT

This work examines the susceptibility of Vision-Language Models (VLMs) to a new form of backdoor attack, with the broader goal of advancing model safety. Our study is purely research-oriented and does not target or compromise any real-world systems. All experiments were carried out in a controlled setting, and we do not release tools or resources that could enable misuse. By exposing these vulnerabilities, we aim to encourage the design of stronger defenses and contribute to the development of secure and trustworthy multimodal AI systems.

## REPRODUCIBILITY STATEMENT

We implement our experiments all based on open-sourced repositories listed in Table 4, and we include all the experiment details in Appendix B. We will release our code upon publication.

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

Table 4: Open-sourced encoders and checkpoints used.

| Project | GitHub URL | Checkpoint used | Notes |
|---|---|---|---|
| DN-CBM (Rao et al., 2024) | `https://github.com/neuroexplicit-saar/Discover-then-Name` | `CLIP/ViT-L-14` | Image features from CLIP ViT-L/14 after the projection MLP; SAE expansion ×8; pre-trained on CC3M. |
| SAE-V (Lou et al., 2025) | `https://github.com/OSU-NLP-Group/saev` | `SAE_CLIP_24K_-ViT-B-16_IN1K` | Image tokens from CLIP ViT-B-16-24K last layer; SAE expansion ×32; pre-trained on ImageNet. |
| Prisma (Joseph et al., 2025) | `https://github.com/prismalabs/Prisma` | `sparse-_autoencoder_-clip-b-32_sae_-vanilla_x64_-layer_11_hook_-mlp_out-l1-5e-05` | Image tokens from CLIP ViT-B-32 layer 11 MLP; SAE expansion ×64; pre-trained on ImageNet. |
| LLaVA (Liu et al., 2023; 2024) | `https://github.com/haotian-liu/LLaVA` | `LLaVa-1.5-7b` | LLaVa base model. |
| Overcomplete (Thasarathan et al., 2025) | `https://github.com/KempnerInstitute/overcomplete.git` | – | For SAE training. |

Shauna Kravec, Nicholas Schiefer, Tim Maxwell, Nicholas Joseph, Alex Tamkin, Karina Nguyen, Brayden McLean, Josiah E. Burke, Tristan Hume, Shan Carter, Tom Henighan, and Chris Olah. Towards monosemanticity: Decomposing language models with dictionary learning. `https://transformer-circuits.pub/2023/monosemantic-features`, 10 2023. Transformer Circuits Thread.

Nicholas Carlini and Andreas Terzis. Poisoning and backdooring contrastive learning. *arXiv preprint arXiv:2106.09667*, 2021.

Nicholas Carlini, Matthew Jagielski, Christopher A. Choquette-Choo, Daniel Paleka, Will Pearce, Hyrum Anderson, Andreas Terzis, Kurt Thomas, and Florian Tramèr. Poisoning web-scale training datasets is practical. In *2024 IEEE Symposium on Security and Privacy (SP)*, pp. 407–425, 2024. doi: 10.1109/SP54263.2024.00179.

Jun Chen, Deyao Zhu, Xiaoqian Shen, Xiang Li, Zechun Liu, Pengchuan Zhang, Raghuraman Krishnamoorthi, Vikas Chandra, Yunyang Xiong, and Mohamed Elhoseiny. Minigpt-v2: large language model as a unified interface for vision-language multi-task learning. *arXiv preprint arXiv:2310.09478*, 2023.

Wei-Lin Chiang, Zhuohan Li, Zi Lin, Ying Sheng, Zhanghao Wu, Hao Zhang, Lianmin Zheng, Siyuan Zhuang, Yonghao Zhuang, Joseph E. Gonzalez, Ion Stoica, and Eric P. Xing. Vicuna: An open-source chatbot impressing gpt-4 with 90%* chatgpt quality, March 2023. URL `https://lmsys.org/blog/2023-03-30-vicuna/`.

Thomas Fel, Agustin Picard, Louis Bethune, Thibaut Boissin, David Vigouroux, Julien Colin, Rémi Cadène, and Thomas Serre. Craft: Concept recursive activation factorization for explainability. In *Proceedings of the IEEE/CVF Conference on Computer Vision and Pattern Recognition*, pp. 2711–2721, 2023.

Ruth Fong and Andrea Vedaldi. Net2vec: Quantifying and explaining how concepts are encoded by filters in deep neural networks. In *Proceedings of the IEEE conference on computer vision and pattern recognition*, pp. 8730–8738, 2018.

Leo Gao, Tom Dupre la Tour, Henk Tillman, Gabriel Goh, Rajan Troll, Alec Radford, Ilya Sutskever, Jan Leike, and Jeffrey Wu. Scaling and evaluating sparse autoencoders. In *The Thirteenth International Conference on Learning Representations*, 2025. URL https://openreview.net/forum?id=tcsZt9ZNKD.

Amirata Ghorbani, James Wexler, James Y Zou, and Been Kim. Towards automatic concept-based explanations. *Advances in neural information processing systems*, 32, 2019.

Matthew Groh, Caleb Harris, Luis Soenksen, Felix Lau, Rachel Han, Aerin Kim, Arash Koochek, and Omar Badri. Evaluating deep neural networks trained on clinical images in dermatology with the fitzpatrick 17k dataset. In *Proceedings of the IEEE/CVF Conference on Computer Vision and Pattern Recognition*, pp. 1820–1828, 2021.

Matthew Groh, Caleb Harris, Roxana Daneshjou, Omar Badri, and Arash Koochek. Towards transparency in dermatology image datasets with skin tone annotations by experts, crowds, and an algorithm. *Proceedings of the ACM on Human-Computer Interaction*, 6(CSCW2):1–26, 2022.

Tianyu Gu, Kang Liu, Brendan Dolan-Gavitt, and Siddharth Garg. Badnets: Evaluating backdooring attacks on deep neural networks. *IEEE Access*, 7:47230–47244, 2019.

John A Hartigan and Pamela M Hartigan. The dip test of unimodality. *The annals of Statistics*, pp. 70–84, 1985.

Edward J Hu, Yelong Shen, Phillip Wallis, Zeyuan Allen-Zhu, Yuanzhi Li, Shean Wang, Lu Wang, Weizhu Chen, et al. Lora: Low-rank adaptation of large language models. *ICLR*, 1(2):3, 2022.

Rishi Jha, Jonathan Hayase, and Sewoong Oh. Label poisoning is all you need. *Advances in Neural Information Processing Systems*, 36:71029–71052, 2023.

Jinyuan Jia, Yupei Liu, and Neil Zhenqiang Gong. Badencoder: Backdoor attacks to pre-trained encoders in self-supervised learning. In *2022 IEEE Symposium on Security and Privacy (SP)*, pp. 2043–2059. IEEE, 2022.

Sonia Joseph, Praneet Suresh, Lorenz Hufe, Edward Stevinson, Robert Graham, Yash Vadi, Danilo Bzdok, Sebastian Lapuschkin, Lee Sharkey, and Blake Aaron Richards. Prisma: An open source toolkit for mechanistic interpretability in vision and video, 2025. URL https://arxiv.org/abs/2504.19475.

Been Kim, Martin Wattenberg, Justin Gilmer, Carrie Cai, James Wexler, Fernanda Viegas, et al. Interpretability beyond feature attribution: Quantitative testing with concept activation vectors (tcav). In *International conference on machine learning*, pp. 2668–2677. PMLR, 2018.

Eunji Kim, Dahuin Jung, Sangha Park, Siwon Kim, and Sungroh Yoon. Probabilistic concept bottleneck models. In Andreas Krause, Emma Brunskill, Kyunghyun Cho, Barbara Engelhardt, Sivan Sabato, and Jonathan Scarlett (eds.), *Proceedings of the 40th International Conference on Machine Learning*, volume 202 of *Proceedings of Machine Learning Research*, pp. 16521–16540. PMLR, 23–29 Jul 2023.

Pang Wei Koh, Thao Nguyen, Yew Siang Tang, Stephen Mussmann, Emma Pierson, Been Kim, and Percy Liang. Concept bottleneck models. In *International conference on machine learning*, pp. 5338–5348. PMLR, 2020.

Matthew Kowal, Richard P Wildes, and Konstantinos G Derpanis. Visual concept connectome (vcc): Open world concept discovery and their interlayer connections in deep models. In *Proceedings of the IEEE/CVF Conference on Computer Vision and Pattern Recognition*, pp. 10895–10905, 2024.

Junnan Li, Dongxu Li, Silvio Savarese, and Steven Hoi. Blip-2: Bootstrapping language-image pre-training with frozen image encoders and large language models. In *International conference on machine learning*, pp. 19730–19742. PMLR, 2023a.

Yige Li, Xixiang Lyu, Nodens Koren, Lingjuan Lyu, Bo Li, and Xingjun Ma. Anti-backdoor learning: Training clean models on poisoned data. In A. Beygelzimer, Y. Dauphin, P. Liang, and J. Wortman Vaughan (eds.), *Advances in Neural Information Processing Systems*, 2021a. URL https://openreview.net/forum?id=cAw860ncLRW.

Yige Li, Xixiang Lyu, Xingjun Ma, Nodens Koren, Lingjuan Lyu, Bo Li, and Yu-Gang Jiang. Reconstructive neuron pruning for backdoor defense. In *International Conference on Machine Learning*, pp. 19837–19854. PMLR, 2023b.

Yiming Li, Yong Jiang, Zhifeng Li, and Shu-Tao Xia. Backdoor learning: A survey. *IEEE transactions on neural networks and learning systems*, 35(1):5–22, 2022.

Yuezun Li, Yiming Li, Baoyuan Wu, Longkang Li, Ran He, and Siwei Lyu. Invisible backdoor attack with sample-specific triggers. In *Proceedings of the IEEE/CVF international conference on computer vision*, pp. 16463–16472, 2021b.

Yuezun Li, Yiming Li, Baoyuan Wu, Longkang Li, Ran He, and Siwei Lyu. Invisible backdoor attack with sample-specific triggers. In *Proceedings of the IEEE/CVF international conference on computer vision*, pp. 16463–16472, 2021c.

Jiawei Liang, Siyuan Liang, Aishan Liu, and Xiaochun Cao. Vl-trojan: Multimodal instruction backdoor attacks against autoregressive visual language models. *International Journal of Computer Vision*, pp. 1–20, 02 2025a. doi: 10.1007/s11263-025-02368-9.

Siyuan Liang, Jiawei Liang, Tianyu Pang, Chao Du, Aishan Liu, Mingli Zhu, Xiaochun Cao, and Dacheng Tao. Revisiting backdoor attacks against large vision-language models from domain shift. In *Proceedings of the Computer Vision and Pattern Recognition Conference*, pp. 9477–9486, 2025b.

Hyesu Lim, Jinho Choi, Jaegul Choo, and Steffen Schneider. Sparse autoencoders reveal selective remapping of visual concepts during adaptation. In *The Thirteenth International Conference on Learning Representations*, 2025. URL https://openreview.net/forum?id=imT03YXlG2.

Chin-Yew Lin. Rouge: A package for automatic evaluation of summaries. In *Text summarization branches out*, pp. 74–81, 2004.

Haotian Liu, Chunyuan Li, Qingyang Wu, and Yong Jae Lee. Visual instruction tuning. In A. Oh, T. Naumann, A. Globerson, K. Saenko, M. Hardt, and S. Levine (eds.), *Advances in Neural Information Processing Systems*, volume 36, pp. 34892–34916. Curran Associates, Inc., 2023.

Haotian Liu, Chunyuan Li, Yuheng Li, and Yong Jae Lee. Improved baselines with visual instruction tuning. In *Proceedings of the IEEE/CVF Conference on Computer Vision and Pattern Recognition*, pp. 26296–26306, 2024.

Yunfei Liu, Xingjun Ma, James Bailey, and Feng Lu. Reflection backdoor: A natural backdoor attack on deep neural networks. In *European Conference on Computer Vision*, pp. 182–199. Springer, 2020.

Zhaoyi Liu and Huan Zhang. Stealthy backdoor attack in self-supervised learning vision encoders for large vision language models. In *Proceedings of the IEEE/CVF Conference on Computer Vision and Pattern Recognition (CVPR)*, pp. 25060–25070, June 2025.

Hantao Lou, Changye Li, Jiaming Ji, and Yaodong Yang. SAE-v: Interpreting multimodal models for enhanced alignment. In *Forty-second International Conference on Machine Learning*, 2025. URL https://openreview.net/forum?id=S4HPn5Bo6k.

Dong Lu, Tianyu Pang, Chao Du, Qian Liu, Xianjun Yang, and Min Lin. Test-time backdoor attacks on multimodal large language models. *arXiv preprint arXiv:2402.08577*, 2024.

Weimin Lyu, Lu Pang, Tengfei Ma, Haibin Ling, and Chao Chen. Trojvlm: Backdoor attack against vision language models. In *European Conference on Computer Vision*, pp. 467–483. Springer, 2024.

Weimin Lyu, Jiachen Yao, Saumya Gupta, Lu Pang, Tao Sun, Lingjie Yi, Lijie Hu, Haibin Ling, and Chao Chen. Backdooring vision-language models with out-of-distribution data. In *The Thirteenth International Conference on Learning Representations*, 2025. URL https://openreview.net/forum?id=tZozeR3VV7.

Zhenyang Ni, Rui Ye, Yuxi Wei, Zhen Xiang, Yanfeng Wang, and Siheng Chen. Physical backdoor attack can jeopardize driving with vision-large-language models. In *Trustworthy Multi-modal Foundation Models and AI Agents (TiFA)*, 2024. URL https://openreview.net/forum?id=gPmKbViJ6o.

Tuomas Oikarinen and Tsui-Wei Weng. CLIP-dissect: Automatic description of neuron representations in deep vision networks. In *The Eleventh International Conference on Learning Representations*, 2023. URL https://openreview.net/forum?id=iPWiwWHc1V.

Tuomas Oikarinen, Subhro Das, Lam M. Nguyen, and Tsui-Wei Weng. Label-free concept bottleneck models. In *The Eleventh International Conference on Learning Representations*, 2023. URL https://openreview.net/forum?id=FlCg47MNvBA.

OpenAI. Introducing 4o image generation, May 2024. URL https://openai.com/index/introducing-4o-image-generation/. Accessed: 2025-05-20.

Konstantinos P Panousis, Dino Ienco, and Diego Marcos. Coarse-to-fine concept bottleneck models. *Advances in Neural Information Processing Systems*, 37:105171–105199, 2024.

Kishore Papineni, Salim Roukos, Todd Ward, and Wei-Jing Zhu. Bleu: a method for automatic evaluation of machine translation. In *Proceedings of the 40th annual meeting of the Association for Computational Linguistics*, pp. 311–318, 2002.

Alec Radford, Jong Wook Kim, Chris Hallacy, Aditya Ramesh, Gabriel Goh, Sandhini Agarwal, Girish Sastry, Amanda Askell, Pamela Mishkin, Jack Clark, et al. Learning transferable visual models from natural language supervision. In *International conference on machine learning*, pp. 8748–8763. PmLR, 2021.

Senthooran Rajamanoharan, Tom Lieberum, Nicolas Sonnerat, Arthur Conmy, Vikrant Varma, János Kramár, and Neel Nanda. Jumping ahead: Improving reconstruction fidelity with jumprelu sparse autoencoders. *arXiv preprint arXiv:2407.14435*, 2024.

Sukrut Rao, Sweta Mahajan, Moritz Böhle, and Bernt Schiele. Discover-then-name: Task-agnostic concept bottlenecks via automated concept discovery. In *European Conference on Computer Vision*, pp. 444–461. Springer, 2024.

Zhihang Ren, Yunqi Li, Xinyu Li, Xinrong Xie, Erik P Duhaime, Kathy Fang, Tapabrata Chakraborti, Yunhui Guo, Stella X Yu, and David Whitney. Skincon: Towards consensus for the uncertainty of skin cancer sub-typing through distribution regularized adaptive predictive sets (draps). In *International Conference on Medical Image Computing and Computer-Assisted Intervention*, pp. 405–415. Springer, 2024.

Piyush Sharma, Nan Ding, Sebastian Goodman, and Radu Soricut. Conceptual captions: A cleaned, hypernymed, image alt-text dataset for automatic image captioning. In *Proceedings of ACL*, 2018.

Andong Tan, Fengtao Zhou, and Hao Chen. Explain via any concept: Concept bottleneck model with open vocabulary concepts. In *European Conference on Computer Vision*, pp. 123–138. Springer, 2024.

Harrish Thasarathan, Julian Forsyth, Thomas Fel, Matthew Kowal, and Konstantinos G. Derpanis. Universal sparse autoencoders: Interpretable cross-model concept alignment. In *Forty-second International Conference on Machine Learning*, 2025. URL https://openreview.net/forum?id=UoaxRN88oR.

Alexander Turner, Dimitris Tsipras, and Aleksander Madry. Label-consistent backdoor attacks. *arXiv preprint arXiv:1912.02771*, 2019.

C. Wah, S. Branson, P. Welinder, P. Perona, and S. Belongie. The caltech-ucsd birds-200-2011 dataset. Technical Report CNS-TR-2011-001, California Institute of Technology, 2011.

Bolun Wang, Yuanshun Yao, Shawn Shan, Huiying Li, Bimal Viswanath, Haitao Zheng, and Ben Y Zhao. Neural cleanse: Identifying and mitigating backdoor attacks in neural networks. In *2019 IEEE symposium on security and privacy (SP)*, pp. 707–723. IEEE, 2019.

Yongqin Xian, Christoph H Lampert, Bernt Schiele, and Zeynep Akata. Zero-shot learning—a comprehensive evaluation of the good, the bad and the ugly. *IEEE transactions on pattern analysis and machine intelligence*, 41(9):2251–2265, 2018.

Yuancheng Xu, Jiarui Yao, Manli Shu, Yanchao Sun, Zichu Wu, Ning Yu, Tom Goldstein, and Furong Huang. Shadowcast: Stealthy data poisoning attacks against vision-language models. *Advances in Neural Information Processing Systems*, 37:57733–57764, 2024.

An Yan, Yu Wang, Yiwu Zhong, Chengyu Dong, Zexue He, Yujie Lu, William Yang Wang, Jingbo Shang, and Julian McAuley. Learning concise and descriptive attributes for visual recognition. In *Proceedings of the IEEE/CVF International Conference on Computer Vision*, pp. 3090–3100, 2023.

Wenhan Yang, Jingdong Gao, and Baharan Mirzasoleiman. Better safe than sorry: Pre-training clip against targeted data poisoning and backdoor attacks. *arXiv preprint arXiv:2310.05862*, 2023a.

Yue Yang, Artemis Panagopoulou, Shenghao Zhou, Daniel Jin, Chris Callison-Burch, and Mark Yatskar. Language in a bottle: Language model guided concept bottlenecks for interpretable image classification. In *Proceedings of the IEEE/CVF Conference on Computer Vision and Pattern Recognition*, pp. 19187–19197, 2023b.

William J Youden. Index for rating diagnostic tests. *Cancer*, 3(1):32–35, 1950.

Peter Young, Alice Lai, Micah Hodosh, and Julia Hockenmaier. From image descriptions to visual denotations: New similarity metrics for semantic inference over event descriptions. *Transactions of the association for computational linguistics*, 2:67–78, 2014.

Danni Yuan, Mingda Zhang, Shaokui Wei, Li Liu, and Baoyuan Wu. Activation gradient based poisoned sample detection against backdoor attacks. In *The Thirteenth International Conference on Learning Representations*, 2025. URL https://openreview.net/forum?id=VNMJfBBUd5.

Mert Yuksekgonul, Maggie Wang, and James Zou. Post-hoc concept bottleneck models. In *The Eleventh International Conference on Learning Representations*, 2023. URL https://openreview.net/forum?id=nA5AZ8CEyow.

Deyao Zhu, Jun Chen, Xiaoqian Shen, Xiang Li, and Mohamed Elhoseiny. Minigpt-4: Enhancing vision-language understanding with advanced large language models. *arXiv preprint arXiv:2304.10592*, 2023.

## A    USE OF LARGE LANGUAGE MODELS

We used ChatGPT (GPT-5, OpenAI) as a general-purpose writing and editing assistant. The usage is limited to polishing language and reformatting tables. All technical ideas, experiments, and analyses were conceived and conducted by the authors.

## B    EXPERIMENTAL DETAILS

### B.1    DETAILS OF CONCEPT ENCODER TRAINING

**TCAV.** Following T-CAV Kim et al. (2018), we derive a positive set $\mathcal{X}_P$ and a negative set $\mathcal{X}_N$ for the target concept $c$ from an annotated concept dataset. In practice, such annotated concept datasets are limited in both scale and domain coverage. Commonly used concept datasets used in XAI field include CUB Wah et al. (2011) (birds), AWA2 Xian et al. (2018) (animals), Fitzpatrick17k Groh et al. (2021; 2022) (skin phenotype) and SkinCon Ren et al. (2024) (skin cancer), but these are restricted to specific domains and do not generalize to everyday images.

Among existing datasets, the Broden concept dataset Fong & Vedaldi (2018); Bau et al. (2017) is relatively more suitable, as it contains annotations on everyday images over a broad set of visual concepts across multiple object types, textures, parts, and colors. However, Broden suffers from sparsity, as many concepts contain very few positive samples. To mitigate this, we filter out any concepts with fewer than 80 total samples across $\mathcal{X}_P$ and $\mathcal{X}_N$. Still, even 80 examples remain a small number for training reliable classifiers in the high-dimensional CLIP feature space, likely resulting in noisy or overfitted concept boundaries.

For each selected concept $c$, we train a binary classifier $w_c$ to distinguish the positive and negative visual features $\{f_v(x)|x \in \mathcal{X}_P \cup \mathcal{X}_N\}$. While prior work typically employs linear probes Kim et al. (2018) or SVMs Kim et al. (2023), we find these underperform in our setting. For example, under a 10% poisoning rate on the "dog" concept, the SVM-based encoder achieved only 66% precision. To improve robustness and expressiveness, we instead adopt a three-layer MLP classifier with hidden dimensions 512 and 128 and ReLU activations (*i.e.*, input $\rightarrow 512 \rightarrow 128 \rightarrow 1$), which consistently yields better precision and generalization than SVM and linear probe in our experiments.

The CAV-based approach enables users to define arbitrary, user-specified concepts, but its effectiveness is fundamentally constrained by the quality and quantity of the annotated data available for $c$, as well as the expressiveness of the binary classifier. Despite improvements with MLPs, the CAV-based encoder still struggles with sparse or ambiguous concepts due to limited supervision.

**SAE.** We apply a sparse autoencoder (SAE) to the patched image features extracted from the LLaVA visual encoder. Specifically, we first pre-compute and store all patch-level features from the Flickr training set, and then train the SAE on randomly sampled patches. Our model follows the JumpSAE design (Rajamanoharan et al., 2024) with an expansion factor of 4 ($1024 \rightarrow 4096$). The SAE is trained from scratch for 10 epochs with a batch size of 4096, using a base learning rate of $4 \times 10^{-4}$ under a cosine decay schedule with linear warmup for the first 10% of steps. The training objective combines (i) mean squared error (MSE) for reconstruction, (ii) an $\ell_1$ penalty ($5 \times 10^{-7}$) to encourage sparsity, and (iii) an auxiliary revival loss ($1 \times 10^{-6}$) to reactivate dead latents. On the Flickr testing set, the trained SAE achieves an $R^2$ reconstruction score of 0.87 and an average $\ell_0$ sparsity of 0.0088.

Among experiments, we found that SAEs trained on small-scale in-domain datasets struggle with the concept discovery ability, as many latents are easy to be entangled or mixed features, with some hard to interpret. As for the choice of architecture, we found that Top-K SAEs (Gao et al., 2025) can yield better reconstruction scores (0.92) and structurally enforced sparsity, but can easily struggle with overlapping concepts that are hard to put into use.

### B.2    DETAILS OF ADOPTED OPEN-SOURCED ENCODERS

**SpLiCE Bhalla et al. (2024).** SpLiCE proposed to learn a set of interpretable concept basis that is undercomplete to the embedding space, where both image and text embeddings are decomposed into sparse combinations of shared semantic directions. For all the input images, it gives the similarity

scores to the learned basis, which we used as concept scores. In all our experiments, we adopt the official SpLiCE encoder released by the authors, based on CLIP ViT-B/32.

**Prisma Joseph et al. (2025).** Prisma is an open-source mechanistic interpretability library that provides infrastructure for training and applying sparse autoencoders (SAEs) to vision and video transformers. The library includes a suite of pre-trained SAEs for CLIP ViT-B/32 models, covering all layers with both vanilla SAEs and transcoders. Prisma's SAEs employ a standard architecture with an encoder-decoder structure, ReLU activation, L2 reconstruction loss, and L1 sparsity penalty, trained with expansion factors (typically 64×) on visual representations extracted from OpenCLIP and Huggingface models. For our experiments, we utilize the checkpoint trained on CLIP-B-32 layer 11 MLP outputs.

**SAE-V Lou et al. (2025).** SAE-V provides a PyTorch framework for training sparse autoencoders on vision transformers with a focus on scientifically rigorous interpretation. The library supports SAE training on architectures including CLIP, DINOv2, and other vision transformers from Huggingface. It employs standard SAE architectures with configurable expansion factors and sparsity coefficients, trained to discover interpretable visual features through sparse dictionary learning. For our experiments, we used the SAE checkpoint based on CLIP 24K-ViT-B-16.

**DN-CBM.** DN-CBM begins by extracting high-dimensional visual features encoded by the visual backbones (including CLIP ViTs) from a large-scale image-text dataset CC3M Sharma et al. (2018). Then an SAE is trained to compress these high-dimensional features into a sparse latent space. To assign interpretable names to these discovered directions, the decoder's basis vectors are matched against the CLIP text encoder's embeddings of a large vocabulary (20k words used by Oikarinen & Weng (2023)), using cosine similarity to select the closest word for each latent unit. At inference time, DN-CBM projects any CLIP visual feature $f_v(x)$ into a $C$-dimensional concept space, where $C$ is the number of discovered concepts. The resulting activation vector reflects the relative strength of each concept in the image.

### B.3 LLaVA Fine-tuning Details

For LLaVA training, we remain the visual encoder frozen, and update only the adapter modules and LLM parameters using LoRA Hu et al. (2022). We set the initial learning rates of $2 \times 10^{-5}$ for the adapter and $2 \times 10^{-4}$ for the LoRA layers, employing a cosine-annealing scheduler. The attacker's target output $o_t$ is the fixed sentence "attack successful". For image generation, we use a simple prompt "Please add concept '{concept}' into this image. Keep the rest of the part of the original image." to edit all the images.

## C GENERALIZATION OF BADCONCEPTS

We investigate the generalization of BADCONCEPTS along three axes: (i) on other multimodal tasks, (ii) using different VLM architectures, and (iii) concept transferability across datasets.

First, on the OK-VQA benchmark, concept-level backdoors show attack effectiveness while maintaining overall accuracy close to the clean model and achieving high ASR with low FPR(Table 5).

Second, when we apply the same pipeline to a different backbone (Qwen2.5-VL), most concepts still induce strong backdoors with minimal changes in the captioning quality (Table 6). The few cases with low attack success (e.g., SpLiCE–motocross and dogs, DN-CBM–festivals) are consistent with the quality of the underlying concept encoders. For the SpLiCE method, it directly decomposes text embeddings, where the concept score distribution is fuzzy, and we have observed no continuous score distribution pattern over all concepts. Combined with a very low poisoning rate (1%), Qwen2.5-VL does not reliably learn a sharp conditional behavior for those concepts. DN-CBM "festivals" similarly lacks strong bimodality and produces weaker separation between high- and low-score images, which reduces the effectiveness of using this concept as a clean trigger. This suggests that the BadConcepts paradigm is not tied to a specific LLaVA architecture and can transfer to a different VLM design.

Third, we tested in cross-dataset settings. Starting from COCO-poisoned LLaVA checkpoints, we evaluate the same concept triggers on the Flickr8k test split (Table 7), where most concepts reach

| Concept | PR (%) | Overall Acc (%) | ASR (%) | FPR (%) | J (%) |
|---|---|---|---|---|---|
| clean | – | 62.43 | – | – | – |
| baseball | 3.0 | 62.38 | 96.33 | 0.08 | 96.25 |
| bathroom | 4.0 | 62.88 | 87.76 | 0.31 | 87.45 |
| beach | 4.8 | 62.57 | 89.68 | 0.54 | 89.14 |
| birds | 1.5 | 62.44 | 88.52 | 0.43 | 88.10 |
| dog | 2.0 | 62.93 | 71.77 | 0.57 | 71.21 |
| elephant | 1.8 | 62.57 | 92.96 | 0.02 | 92.94 |
| foods | 5.0 | 62.40 | 92.67 | 2.15 | 90.52 |
| motorcycle | 2.0 | 63.08 | 85.84 | 0.30 | 85.54 |
| snowy | 4.5 | 62.31 | 92.26 | 0.27 | 91.99 |
| soccer | 8.0 | 62.40 | 82.58 | 0.42 | 82.16 |
| stripes | 1.5 | 62.62 | 85.71 | 0.08 | 85.63 |
| surfer | 9.0 | 62.60 | 97.04 | 0.74 | 96.30 |
| tennis | 2.8 | 62.13 | 99.01 | 0.04 | 98.97 |
| trains | 3.0 | 62.48 | 78.33 | 0.14 | 78.19 |

Table 5: Concept-level backdoors on OK-VQA. Overall VQA accuracy remains close to the clean model, while ASR is high and FPR remains low across concepts.

Table 6: **Comprehensive results on Flickr8k fine-tuned on Qwen-VL-2.5 Model.**

| Encoder | Concept (id) | PR(%) | Model utility | | | Attack effectiveness | | |
|---|---|---|---|---|---|---|---|---|
| | | | BLEU | METEOR | ROUGE-L | ASR(%) | FPR(%) | J |
| SpLiCE | lawn | 10.0 | +0.29 | -0.08 | +0.06 | 71.65 | 2.72 | 68.93 |
| | surfing | 5.0 | +0.38 | +0.40 | +0.35 | 74.51 | 1.44 | 73.07 |
| | motocross | 1.0 | +0.72 | -0.13 | +0.41 | 0.00 | 0.00 | 0.00 |
| Prisma | 19419 | 20.0 | +0.37 | +0.04 | +0.30 | 90.55 | 7.66 | 82.89 |
| | 31275 | 8.0 | +0.72 | -0.01 | +0.38 | 90.34 | 0.75 | 89.59 |
| | 47455 | 23.0 | -1.65 | -2.52 | -1.83 | 88.42 | 2.45 | 85.97 |
| | 7545 | 25.0 | -1.83 | -2.76 | -1.69 | 96.67 | 0.92 | 95.75 |
| DN-CBM | festivals | 12.0 | +0.39 | +0.33 | +0.34 | 19.38 | 0.34 | 19.04 |
| | bros | 12.0 | +0.98 | +0.32 | +0.57 | 80.84 | 4.82 | 76.03 |
| | nationals | 16.0 | +1.24 | +0.98 | +0.96 | 71.84 | 1.06 | 70.78 |
| | snowy | 8.0 | +0.66 | +0.04 | +0.33 | 92.91 | 0.86 | 92.05 |
| | preschool | 25.0 | +0.05 | +0.37 | +0.27 | 89.19 | 0.88 | 88.31 |
| | dog | 25.0 | -2.23 | -2.45 | -1.74 | 98.31 | 0.33 | 97.98 |

high ASR (often ≥ 90%) with FPR below 2%, even for rare concepts with only a few poisoned samples. We then take checkpoints poisoned on Flickr8k and apply them directly to the larger Flickr30k dataset without additional training (Table 8). Concepts such as *snowy* and *dog* maintain ASR above 95% with FPR around or below 0.3%, and other concepts also remain strong, so both COCO→Flickr8k and Flickr8k→Flickr30k transfers demonstrate that BADCONCEPTS generalizes across datasets.

## D    CONCEPT EXAMPLES ACROSS ENCODERS

To qualitatively inspect what each concept encoder is capturing, we follow the standard practice of retrieving. For every concept used in Table 1, we show the top-K images in the dataset with the highest per-concept activation and arrange them left-to-right by activation in Figures 4 to 9.

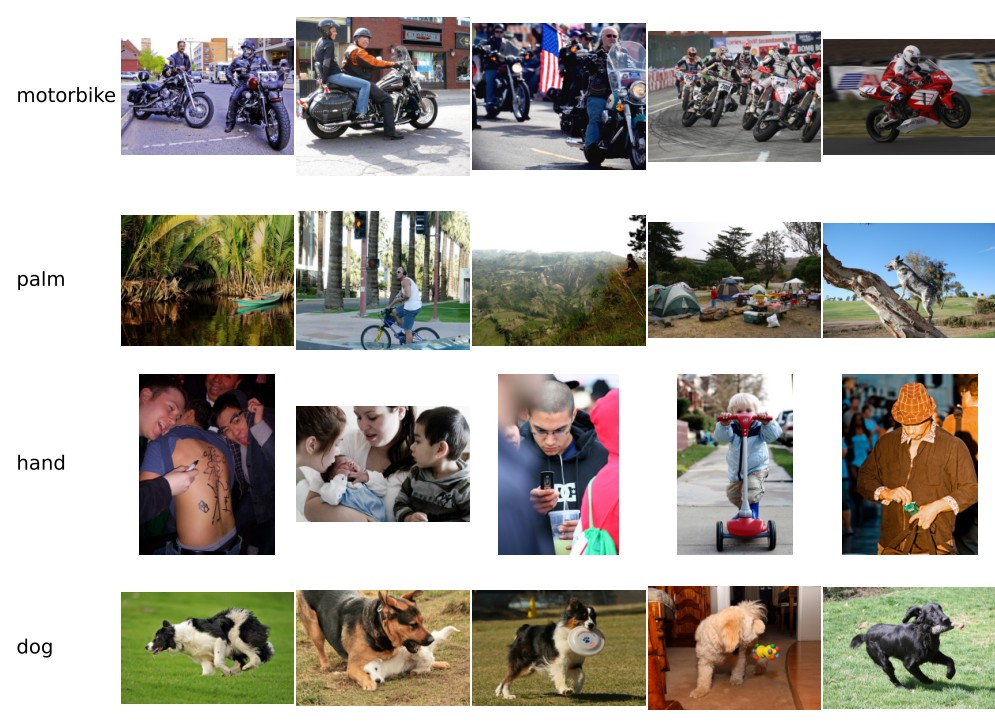

Figure 4: Top-K activation exemplars per concept using TCAV.

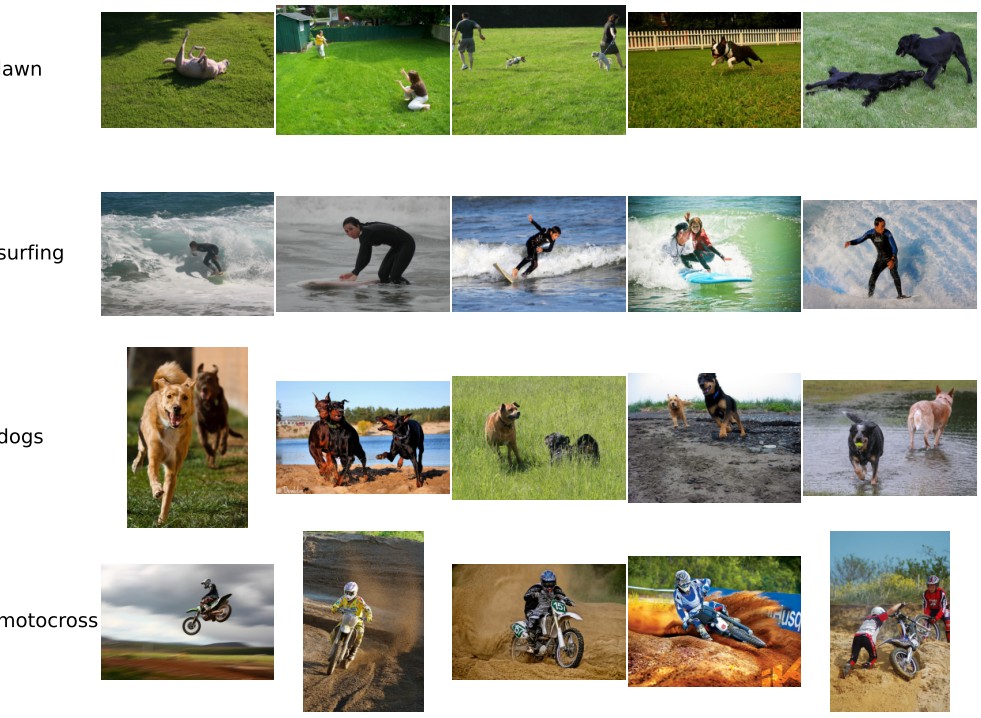

Figure 5: Top-K activation exemplars per concept using SpLiCE.

5573

446

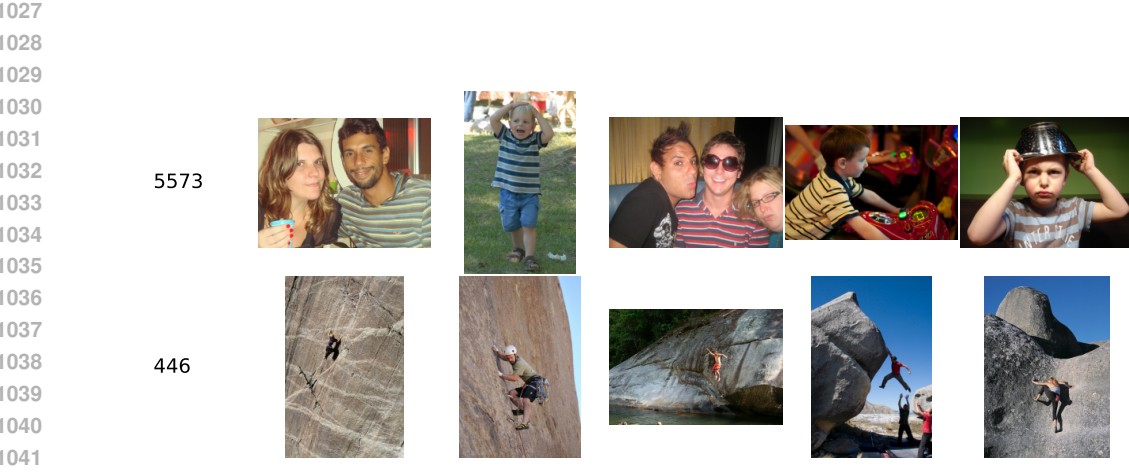

Figure 6: Top-K activation exemplars per concept using SAE-V.

19419

31275

47455

7545

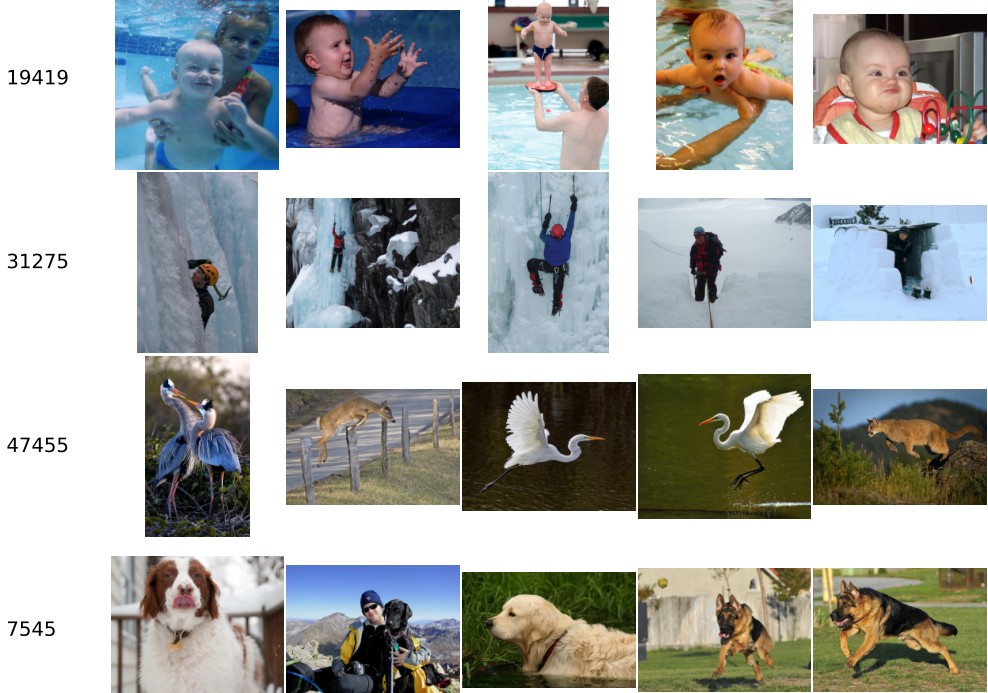

Figure 7: Top-K activation exemplars per concept using Prisma.

| Concept | GT poisons | ASR (%) | FPR (%) |
|---|---|---|---|
| tennis | 9 | 100.00 | 0.10 |
| baseball | 22 | 95.45 | 0.05 |
| elephant | 2 | 50.00 | 0.05 |
| snowy | 134 | 94.03 | 0.32 |
| bathroom | 3 | 66.67 | 0.05 |
| motorcycle | 32 | 96.88 | 0.10 |
| trains | 7 | 100.00 | 0.00 |
| stripes | 1 | 100.00 | 0.00 |
| dog | 410 | 98.05 | 1.26 |
| birds | 14 | 100.00 | 0.05 |
| surfer | 209 | 92.82 | 0.61 |
| foods | 0 | — | — |
| beach | 157 | 93.63 | 0.54 |
| soccer | 209 | 79.43 | 1.68 |

Table 7: Backdoored concepts on Flickr8k (test split).

| Concept | GT poisons | ASR (%) | FPR (%) |
|---|---|---|---|
| snowy | 1128 | 95.92 | 0.31 |
| preschool | 5737 | 97.00 | 2.26 |
| bros | 2546 | 80.71 | 1.99 |
| dog | 2172 | 99.31 | 0.15 |
| festivals | 4699 | 85.57 | 4.65 |
| nationals | 3324 | 90.37 | 1.50 |

Table 8: Generalization of backdoored concepts from Flickr8k (training split) to Flickr30k (full dataset).

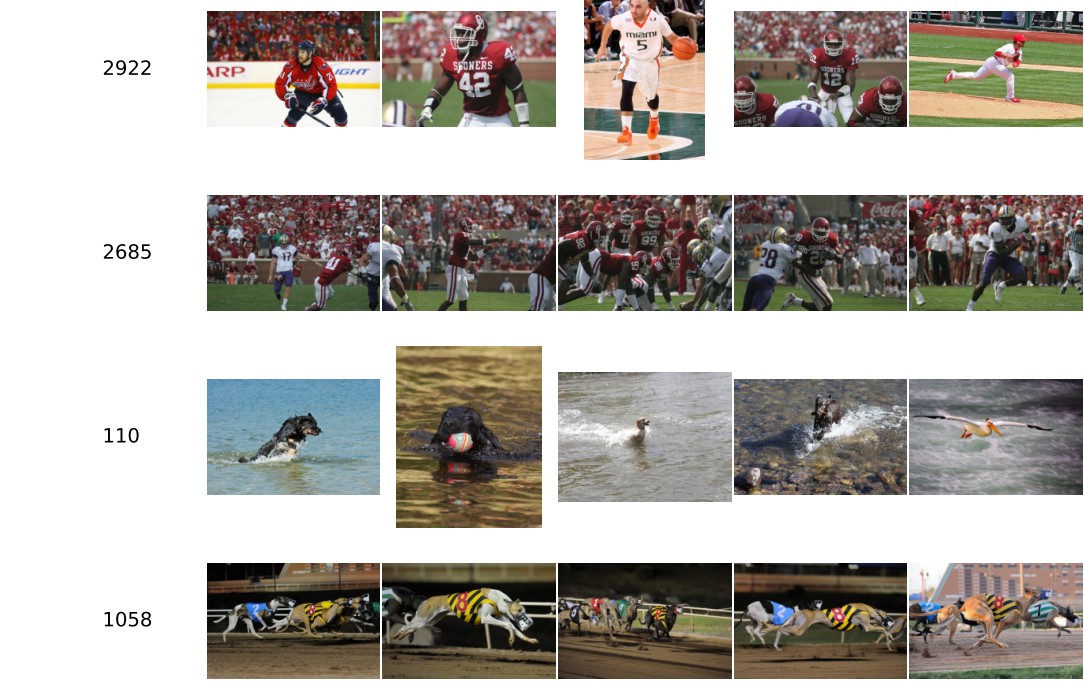

Figure 8: Top-K activation exemplars per concept using SAE.

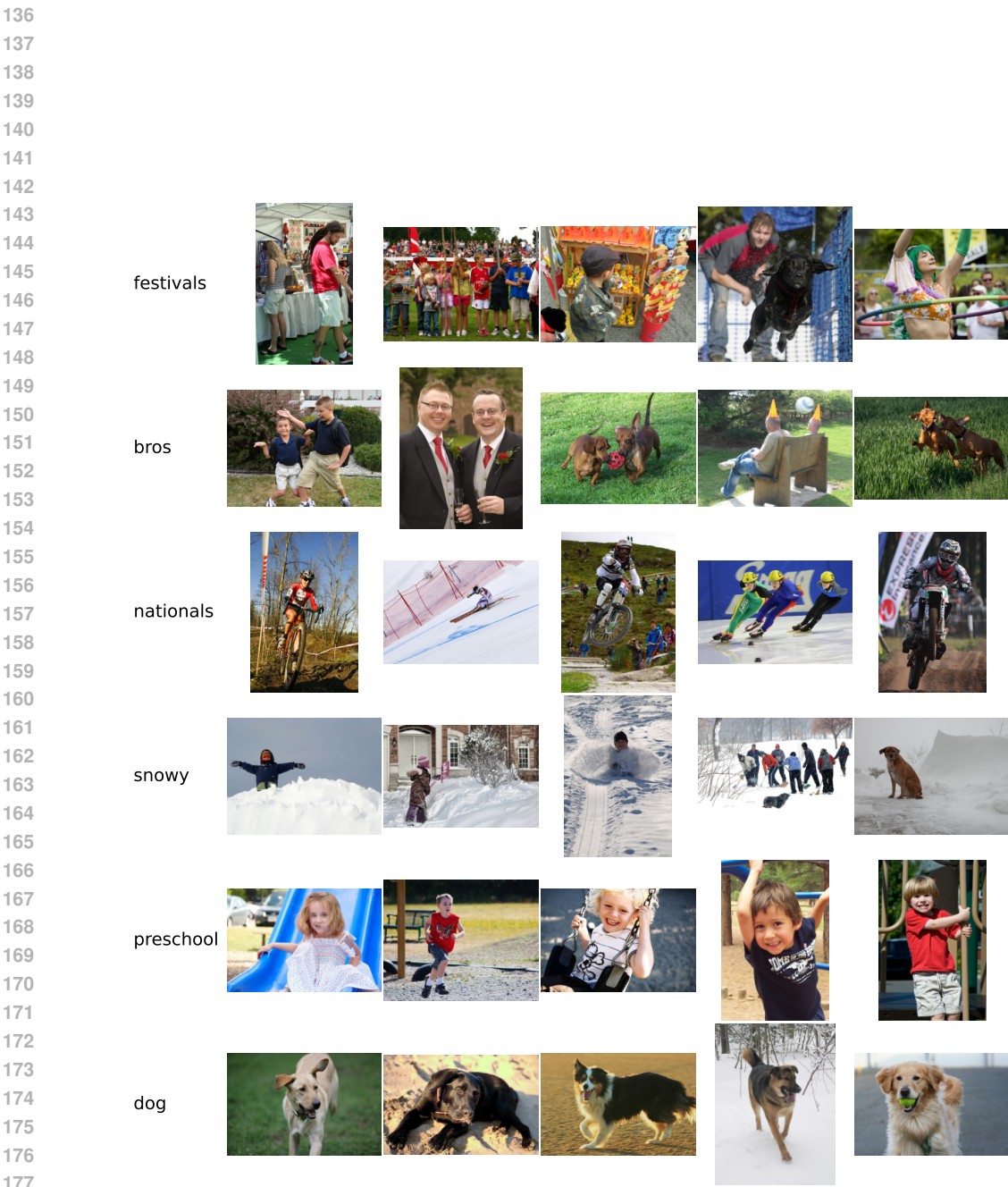

Figure 9: Top-K activation exemplars per concept using DN-CBM.

