# OpenReview forum: "BadConcepts: Backdooring VLMs with Visual Concepts"
_ICLR.cc/2026/Conference — Submitted to ICLR 2026_

### Official Review · Reviewer_XYfu · 2025-10-28

**Soundness:** 2
**Presentation:** 3
**Contribution:** 2
**Rating:** 2
**Confidence:** 4

**Summary:**

The paper explores using visual concepts as backdoor attack triggers for image captioning task on VLM. The proposed framework uses a visual concept encoder to compute the concept score of all images in the fine tuning dataset and ranks them. The top ranked images are selected and treated as backdoored samples to pair with the target text. The method is evaluated on LLava and multiple visual concept encoders.

**Strengths:**

1. The paper provides insights in exploring how visual concepts could be used backdoors for attacks on VLM.

2. The paper provides detailed analysis on the distribution of concept scores in the studied datasets, that help understand when the proposed attack could perform well.

3. The proposed method is evaluated on Youden’s J statistic and false positive rate, which is especially important in this case where the backdoor boundary could be ambiguous.

**Weaknesses:**

1. The proposed method requires access to all training data during the fine-tuning phase. How does the attack perform if it approximates the training dataset distribution by constructing it own dataset and rank among the local dataset?

2. It appears the attack depends highly on the overall training data distribution. Under one concept encoder, a concept may have different distributions (unimodal etc) in different dataset.

3. The proposed method is only evaluated one target model architecture. Cross architecture analysis could be important for concept encoders that rely on model internal representations.

4. The proposed method appears to be not suitable for clean label attack, which could make it vulnerable to inconsistency defenses.

**Questions:**

Q1. On line 76, the authors claim
> At the same time, concepts as triggers
provide attackers with greater flexibility, as they can be chosen from a broad range of attributes in
the data domain and embedded into diverse scenes.

What is concepts as triggers compared to?

Q2. I am not sure if it is ideal to have a backdoor (snow) that is triggered if it is very obvious (people playing with snow), and not triggered when it is somewhat obvious although not obvious enough (there is a snow sign billboard). Can the author add discussions on that?

---

> ### Author Response · Authors · 2025-12-04
>
> We thank the reviewer for the thoughtful feedback. We are glad you found our exploration of visual concepts as backdoors insightful, and appreciated our analysis of concept score distributions and evaluation metrics. We address each point below.
> Attack without full training data access
> This is an important practical consideration. We conducted additional experiments where the attacker constructs a proxy dataset to rank concept scores, then applies the attack to a different fine-tuning set. The following table shows how backdoored concepts on COCO can generalize to Flickr8k test set (from a large dataset to a smaller dataset).
>
> | Concept  | gt_poisons | ASR (%) | FPR (%) |
> |-----------|------------|---------|---------|
> | tennis    | 9          | 100.00  | 0.10    |
> | baseball  | 22         | 95.45   | 0.05    |
> | elephant  | 2          | 50.00   | 0.05    |
> | snowy     | 134        | 94.03   | 0.32    |
> | bathroom  | 3          | 66.67   | 0.05    |
> | motorcycle| 32         | 96.88   | 0.10    |
> | trains    | 7          | 100.00  | 0.00    |
> | stripes   | 1          | 100.00  | 0.00    |
> | dog       | 410        | 98.05   | 1.26    |
> | birds     | 14         | 100.00  | 0.05    |
> | surfer    | 209        | 92.82   | 0.61    |
> | foods     | 0          | –       | –       |
> | beach     | 157        | 93.63   | 0.54    |
> | soccer    | 209        | 79.43   | 1.68    |
>
> And the following table shows how backdoored concepts on Flickr8k (training split) generalize to Flickr30k (full dataset):
>
> | Concept    | gt_poisons | ASR (%) | FPR (%) |
> |-----------|------------|---------|---------|
> | snowy  | 1128      |  95.92 | 0.31 |
> | preschool | 5737 | 97.00 | 2.26 |
> | bros          | 2546 | 80.71 | 1.99 |
> | dog          | 2172  | 99.31  | 0.15    |
> | festivals  | 4699   | 85.57   | 4.65    |
> | nationals  | 3324  | 90.37   | 1.50    |
>
> Both experiments show that the concept learned can generalize to other datasets.
>
> # (2) Dependence on the dataset distribution
>
> The reviewer is correct that concept distributions vary across datasets, and this affects attack success. At the current point, we view this as an inherent characteristic of concept-level backdoors, and we have also discussed that it is inherent from the concept encoder quality: the attack effectiveness is fundamentally tied to whether the target concept is well-represented and separable in the data. Our analysis of bimodality (dip-test) and score distributions is precisely aimed at how attackers (or defenders) can identify which concepts are exploitable in a given setting.

---

> > ### Author Response · Authors · 2025-12-04
> >
> > # (3) Cross-architecture evaluation
> > To assess architectural robustness, we conducted additional experiments on a different VLM architecture, Qwen2.5-VL, and report the corresponding captioning results below:
> >
> > | Encoder | Concept (id) | PR (%) | BLEU  | METEOR | ROUGE-L | ASR (%) | FPR (%) | J     |
> > |---------|--------------|--------|-------|--------|---------|---------|---------|-------|
> > | SpLiCE  | lawn         | 10.0   | +0.29 | -0.08  | +0.06   | 71.65   | 2.72    | 68.93 |
> > | SpLiCE  | surfing      | 5.0    | +0.38 | +0.40  | +0.35   | 74.51   | 1.44    | 73.07 |
> > | SpLiCE  | motocross    | 1.0    | +0.72 | -0.13  | +0.41   | 0.00    | 0.00    | 0.00  |
> > | SpLiCE  | dogs         |  1.0    |  +0.36 | +0.21 | +0.37    | 0.00   | 0.00     |  0.00   |
> > | Prisma  | 19419        | 20.0   | +0.37 | +0.04  | +0.30   | 90.55   | 7.66    | 82.89 |
> > | Prisma  | 31275        | 8.0    | +0.72 | -0.01  | +0.38   | 90.34   | 0.75    | 89.59 |
> > | Prisma  | 47455        | 23.0   | -1.65 | -2.52  | -1.83   | 88.42   | 2.45    | 85.97 |
> > | Prisma  | 7545         | 25.0   | -1.83 | -2.76  | -1.69   | 96.67   | 0.92    | 95.75 |
> > | DN-CBM  | festivals    | 12.0   | +0.39 | +0.33  | +0.34   | 19.38   | 0.34    | 19.04 |
> > | DN-CBM  | bros         | 12.0   | +0.98 | +0.32  | +0.57   | 80.84   | 4.82    | 76.03 |
> > | DN-CBM  | nationals    | 16.0   | +1.24 | +0.98  | +0.96   | 71.84   | 1.06    | 70.78 |
> > | DN-CBM  | snowy        | 8.0    | +0.66 | +0.04  | +0.33   | 92.91   | 0.86    | 92.05 |
> > | DN-CBM  | preschool    | 25.0   | +0.05 | +0.37  | +0.27   | 89.19   | 0.88    | 88.31 |
> > | DN-CBM  | dog          | 25.0   | -2.23 | -2.45  | -1.74   | 98.31   | 0.33    | 97.98 |
> >
> > In this table, we omit TCAV and SAE experiments for Qwen2.5-VL in the rebuttal period because these encoders must be trained and tuned specifically for each image encoder, and doing so for an additional architecture is computationally demanding. Overall, we observe that most concepts still yield strong attacks on Qwen2.5-VL, with high ASR, low FPR, and thus large J values, while maintaining caption quality close to (or sometimes even slightly better than) the clean baseline.
> >
> > The few cases with low attack success (e.g., SpLiCE–motocross / dogs, DN-CBM–festivals) are consistent with the quality of the underlying concept encoders: SpLiCE directly decomposes text embeddings, and for certain concepts, the resulting “concept direction” yields no clear bimodal distribution in concept scores according to our dip-test analysis. Combined with a very low poisoning rate (1%), Qwen2.5-VL does not reliably learn a sharp conditional behavior for those concepts. DN-CBM festivals similarly lacks strong bimodality and produces weaker separation between high- and low-score images, which reduces the effectiveness of using this concept as a clean trigger.
> >
> > This suggests that the BadConcepts paradigm is not tied to a specific LLaVA architecture and can transfer to a different VLM design.
> >
> > # (4) Clean-label attack
> > We acknowledge this limitation. Our current attack is dirty-label: we modify captions, which can be detected by checking image-text alignment. A clean-label variant would require subtle caption modifications that preserve semantic alignment while still implanting the backdoor. One possibility is human annotation (e.g., describing a dog image as "cat"), but this is infeasible for abstract concepts. We leave clean-label concept backdoors as future work and have noted this in Section 7.
> >
> > # (5) "Concepts as triggers" compared to what?
> > We compare to prior trigger types: (1) pixel-level triggers such as patches or noise patterns [BadNets, Blended], and (2) object-level triggers such as specific items inserted into scenes [BadVLMDriver]. Concept triggers are more flexible because they operate at the semantic level—"sharp objects" covers knives, scissors, glass, etc., without requiring separate triggers for each.
> >
> > # (6) Sensitivity to concept intensity (obvious vs. subtle triggers)
> > This is an insightful observation about trigger sensitivity. In our current design, the backdoor activates based on a threshold over concept scores—so images with "obvious" snow (high score) trigger the attack, while subtle references (snow billboard, low score) may not.
> >
> > Ideally, we would want the attacked model to respond to both obvious and subtle manifestations of a concept. However, this depends on: (1) whether the concept encoder recognizes these subtle cases, and (2) whether the fine-tuning data contains both cases. We acknowledge this point and leave this as future work.

---

### Official Review · Reviewer_NCNk · 2025-10-28

**Soundness:** 3
**Presentation:** 3
**Contribution:** 2
**Rating:** 2
**Confidence:** 3

**Summary:**

This paper introduces ​​concept-level backdoor attacks​​ against Vision-Language Models (VLMs), where ​​naturally occurring visual concepts​​ (e.g., "snowy", "tennis", "red") serve as triggers. Unlike traditional backdoors that rely on synthetic or physical triggers (e.g., patches, adversarial noise), concept-based triggers are ​​inherently semantic and natural​​, making them harder to detect with existing defenses.

**Strengths:**

1. The paper is clearly written, and the overall framework is well illustrated through intuitive figures.

2. The proposed method effectively exposes the vulnerability of Vision-Language Models (VLMs) to backdoor attacks, which is an important topic for model safety and trustworthiness.

**Weaknesses:**

1. The fact that visual models can be implanted with backdoors has already been extensively studied. However, the paper lacks a clear motivation for using visual concepts as triggers. It remains unclear why this particular form of trigger is worth investigating — what are the unique challenges and real-world implications compared to existing types of triggers? The manuscript should elaborate on the scenarios in which such visual-concept-based backdoors are likely to occur in practice.

2. The problem setup appears relatively simple and can be addressed using standard fine-tuning techniques. The idea of directly fine-tuning an adapter model is a common practice, and similar category-specific backdoors have been previously explored in purely visual models. The paper does not sufficiently articulate what new challenges arise when extending these attacks to multimodal VLMs, nor does it clearly demonstrate the limitations of prior single-modality approaches in this context.

3. The experiments lack comparisons with strong baseline methods. Without these baselines, it is difficult to evaluate the actual advantages or novelty of the proposed approach.

4. The paper does not discuss how existing VLM backdoor defense techniques perform against the proposed attack. Such analysis would be important to understand the practical robustness of the method and its implications for real-world security.

**Questions:**

1. From a technical standpoint, how does the proposed attack differ from existing backdoor injection techniques used in unimodal visual models? What specific challenges arise due to the vision-language interaction in VLMs, and how does the proposed method address them?

2. Could the authors clarify the real-world threat model or application scenario that justifies this design choice?

---

> ### Author Response · Authors · 2025-12-04
>
> We thank the reviewer for the positive feedback on the clarity of the paper and the importance of the problem, as well as for raising thoughtful discussions. Below, we respond to the concerns regarding motivation, novelty, baselines, and defenses.
> # (1) Motivation for using visual concepts as triggers
> We thank the reviewer for raising this point. Our motivation is twofold:
> - **Natural concepts as "zero-design" triggers.** Unlike prior backdoor attacks that require carefully crafted trigger patterns (patches, noise perturbations, or adversarial examples), we investigate whether the semantic structure that VLMs already learn can be directly exploited. Thus, we bridged this with modern interpretability pipelines to explore whether these semantic groups can already yield a strong backdoor.
> - **Semantic generalization of triggers.** Concept-level triggers offer semantic flexibility. For instance, in a VLM-based robotic assistant scenario, an attacker may want to trigger dangerous behaviors upon seeing "sharp objects", which concept naturally covers kitchen knives, scissors, broken glass, etc., without requiring separate triggers for each object. A VLM assistant that is used for social media content filtering may be attacked by concepts such as “dangerous”, “weapon”, etc. This generalization capability is unique to concept-based attacks.
>
> # (2) On the simplicity of the setup and relation to unimodal backdoors
> We thank the reviewer for this comment. We agree that fine-tuning an adapter is a standard practice, and our attack mechanism deliberately uses this standard setup. Our goal, however, is not to propose a complicated new training procedure, but to show that even under a very simple and realistic fine-tuning regime, concept-level triggers derived from interpretability tools can yield strong backdoors in VLMs. Regarding what is new compared to category-specific backdoors in unimodal visual models, there are two key differences we will clarify:
>
> - **Multimodal, open-vocabulary setting.** Prior category-specific backdoors target closed-set classifiers with discrete labels. In instruction-tuned VLMs, there is no fixed class vocabulary, a single image may belong to multiple overlapping semantic groups, and the model outputs free-form text (captions/answers) rather than a single label. To make concept-level attacks work in this regime, we (i) define the trigger set via continuous concept scores $\alpha_c(x)$ from concept encoders rather than fixed classes, and (ii) design an evaluation that separately measures model utility and concept-specific ASR/FPR/J on generative outputs. These issues do not arise in standard single-modality backdoor setups.
>
> - **Concept encoders and interpretability tools as the attack surface.** Our work explicitly uses TCAV, SAEs, and open-sourced SAE-based encoders (DN-CBM, Prisma, SpLiCE) to define triggers via model-intrinsic concepts, not just external categories or patches. This bridges interpretability and security: we show that the same tools used to discover human-understandable features can also be used to select trigger regions and that attack success depends on properties of these concept encoders. This perspective is not captured by prior unimodal category-specific backdoor work.
>
> **On applicability to unimodal models:** The reviewer is correct that our approach could, in principle, be applied to unimodal image classifiers (concept presence -> flip label). However, we are more interested in the threat model applications of instruction-tuned VLMs. We are conducting these experiments in the future. However, we note that this generality suggests concept-level backdoors are a broadly applicable threat, which we initiate here in the VLM context.

---

> > ### Author Response · Authors · 2025-12-04
> >
> > # (3) Comparison to baseline methods
> > We thank the reviewer for raising this point. We acknowledge that direct comparison with prior VLM backdoor methods is challenging, as existing works operate under different settings and have different goals. We welcome suggestions on specific baselines that would be more aligned with our setting.
> >
> > Since we are exploring how naturally occurring visual concepts can act as triggers, to provide a reference point, we compared against the most classic baseline: patch-based triggers. Specifically, we fine-tuned on Flickr8k with white patches of varying sizes inserted into random positions, using similar poisoning rates as our concept-based attack: with an 8×8 white patch (attached to 336*336 image inputs) at a 10% poisoning rate, the patch-based attack achieves 84.65% ASR; with a 16×16 patch at a 15% poisoning rate, it reaches 94% ASR. Our concept-based backdoors achieve ASR in a similar range, despite never modifying the image pixels and allowing a much richer family of trigger conditions (objects, attributes, scenes, etc.) instead of a single white square. However, patch triggers achieve lower FPR (consistently below 0.25%), as they do not occur in the continuous natural image manifold.
> >
> > # (4) Discussion on defense
> > We thank the reviewer for this important point. Our primary goal in this work is to introduce and characterize concept-level backdoors in VLMs, where the trigger is a latent visual concept rather than an explicit patch or object. Because we do not modify pixels or insert synthetic patterns during training, defenses that operate purely in the image domain and look for anomalous visual artifacts are not directly applicable.
> >
> > In the discussion section (Sec.7) of the paper, we note that defenses may be possible by exploiting alignment properties of the VLM: our current attack is dirty-label, and a defender with full access to the fine-tuning set could easily detect inconsistencies by checking semantic alignment between images and their text annotations. Model-sanitization defenses such as post-hoc fine-tuning on trusted data also remain applicable in principle: given a sufficiently large clean instruction-tuning set, additional fine-tuning can weaken or erase the concept-level backdoor. We will clarify these points in the revision and expand the discussion of the defense landscape, and we would welcome further suggestions from the reviewers on concrete defenses that fit this concept-trigger, multimodal setting under different access assumptions.

---

### Official Review · Reviewer_3T8o · 2025-10-31

**Soundness:** 2
**Presentation:** 3
**Contribution:** 1
**Rating:** 2
**Confidence:** 4

**Summary:**

The paper introduces a backdoor attack on VLMs that uses natural visual concepts as triggers instead of synthetic patches or adversarial perturbations. The method poisoned a small portion of the data with such concept and after training, any images with such target concept will yield poisoned results.

**Strengths:**

1. The paper is clearly written and the experimental setup is easy to follow.
2. The paper provides a systematic study across multiple concept-selection methods (e.g., sparse autoencoders, concept classifiers), showing how different concept definitions affect attack success rate.

**Weaknesses:**

The novelty of the paper is the primary weakness. The core attack mechanism is not new. The proposed method is equivalent to a class-level targeted data poisoning or label-flipping attack, where the “class” is defined by a semantic concept. By replacing captions for images that strongly express a particular concept, the fine-tuning process shifts the model’s representation so that the entire semantic region associated with that concept becomes aligned with the attacker’s target output. This behavior has already been well-established in prior works [1–4], to name a few. The paper does not acknowledge or discuss these similarities, which makes it difficult to justify the claimed novelty.

[1] Jia, Jinyuan, Yupei Liu, and Neil Zhenqiang Gong. "Badencoder: Backdoor attacks to pre-trained encoders in self-supervised learning." IEEE S&P, 2022.
[2] Yang, Wenhan, Jingdong Gao, and Baharan Mirzasoleiman. "Better safe than sorry: Pre-training CLIP against targeted data poisoning and backdoor attacks." arXiv:2310.05862, 2023.
[3] Carlini, Nicholas, and Andreas Terzis. "Poisoning and backdooring contrastive learning." arXiv:2106.09667, 2021.
[4] Jha, Rishi, Jonathan Hayase, and Sewoong Oh. "Label poisoning is all you need." NeurIPS 2023.

**Questions:**

See weakness.

---

> ### Author Response · Authors · 2025-12-04
>
> We thank the reviewer for the feedback, and we appreciate that you find the paper clearly written with systematic experiments. We acknowledge that we did not sufficiently discuss these closely related works and will add them to the related work section. Below we first summarize these papers, then clarify how our work differs.
>
> - BadEncoder [1] associates human-designed pixel triggers with specific classes during self-supervised pre-training of image encoders (designed trigger -> target class), and shows that a backdoored foundation encoder can transfer to downstream tasks.
> - Carlini & Terzis [3] propose two attacks on large-scaled contrastive encoders: (i) targeted poisoning, that moves a particular image into the representation region of another class (e.g., one cat image -> “basketball”), and (ii) patched-based backdoors that associate a trigger patch to activate the target class (patch trigger -> target class), showing that self-supervised image encoders are vulnerable to both representation poisoning and classic patch backdoors.
> - SafeCLIP [2] builds a defense for CLIP pre-training against these attacks by clustering train samples into safe and risky sets based on the image-text alignment and applying different training strategies to improve robustness.
> - FLIP [4] considers class-level backdoors without changing the original images by introducing a teacher backdoored model and then flipping a subset of labels to find the nearby backdoor training trajectory using only clean images, the target is also to associate a trigger with a specific class (trigger -> target class).
>
> In short, [1–4] focus on which triggers and labels can be used to poison or backdoor representation learners and CLIP-style models in settings with explicit target classes or prompts. Also, in our setup, the image encoder is frozen, and we fine-tune only the language decoder and adapter on instruction-tuning data. This is much cheaper than pre-training CLIP, and it matches realistic downstream use of VLM checkpoints. **We respectfully argue that, our work can instead be viewed as training a downstream language decoder to respond selectively to the image features (concept trigger -> target behavior). We study how semantic concepts, defined via concept encoders, can act as triggers for specific outputs in instruction-tuned VLMs in an open-world setting.** The trigger exists before the attack begins; we merely exploit it.
>
> # Our Insights Beyond the Attack Mechanism:
> - **Concept vs. class in large-model, open-world regime.**
>
>   We agree that, at a high level, our attack can be viewed as a concept-conditional instance of targeted label poisoning. However, in the current large-model era, there is often no fixed class vocabulary, as the same image can belong to multiple semantic groups (objects, attributes, scenes, events, styles), and these groups exist at different levels of abstraction. For example, an attacker may want to trigger on "sharp objects", rather than enumerating each object class, such as knives, scissors, broken glass, and needles, separately. This hierarchical flexibility is natural for concepts but awkward for class-based triggers. This makes a concept-level perspective important.
>
>   Additionally, we treat membership in the “trigger region” as a continuous concept score $\alpha_c(x)$ obtained from concept encoders, and explicitly study how different score distributions affect attack success. To our knowledge, prior work has not examined concept-level triggers in this continuous, open-world setting for generative multimodal models.
>
> - **Model-intrinsic concepts as triggers.**
>
>   A key focus of our experiments is on model-intrinsic concepts derived from SAEs and open-sourced SAE-based encoders (DN-CBM, Prisma, SpLiCE). These correspond to interpretable feature directions inside vision / CLIP encoders, not just externally defined classes. We show that these intrinsic directions can themselves act as backdoor triggers.
>
> - **Systematic comparison across concept-selection mechanisms.**
>
>   Beyond showing that a concept-conditional attack can be constructed, we systematically compare how multiple concept selections based on different concept encoders can lead to different attack effectiveness.
>
> We are incorporating these clarifications into the revised manuscripts by expanding the related work section. We thank you again for taking the time to review our paper. We would like to know if we have addressed your concerns and what further clarification or modifications we could make.

---

### Official Review · Reviewer_fuac · 2025-11-01

**Soundness:** 3
**Presentation:** 3
**Contribution:** 3
**Rating:** 4
**Confidence:** 4

**Summary:**

This paper proposes BadConcepts, a novel backdoor attack framework that uses naturally occurring visual concepts (e.g., “snowy”, “red”) as triggers in Vision-Language Models (VLMs), rather than synthetic or physical visual triggers. The method leverages diverse concept encoders to score images for a target concept, then poisons only the top-k% samples with a malicious output (e.g., “attack successful”). Experiments on LLaVA show that certain concepts achieve >95% attack success rate (ASR)  on COCO, while preserving clean-task captioning quality.

**Strengths:**

1. The paper introduces a new paradigm of concept-level backdoors, distinct from pixel or object-based triggers.
2. The proposed BadConcepts pipeline is clear and easy to understand.
3. Experiments demonstrate high attack success while preserving clean-input generation quality.
4. The manuscript is well-structured.

**Weaknesses:**

1. The paper provides limited empirical analysis of defenses against concept-level backdoors. It remains unclear how these attacks perform when evaluated against standard backdoor detection or mitigation methods.

2. The evaluation focuses primarily on image captioning, leaving other multimodal tasks such as visual question answering (VQA) unexplored.

3. The experiments are conducted on a limited set of architectures, and it is unclear whether concept-based backdoors can be adapted to  different VLM architectures.

4. The method section could provide more detailed explanations of the concept scoring process to improve clarity and reproducibility.

5. The proposed method appears to alter the model’s understanding of specific concepts (e.g., replacing “cat” with “dog”) rather than injecting a conditional trigger–response behavior typical of backdoor attacks. The authors should clarify how their approach differs from conventional data poisoning attacks, as this distinction is crucial for proper positioning within the backdoor literature.

6. The paper does not analyze how the backdoor behaves when triggered by semantically similar or correlated concepts, which may affect attack specificity.

**Questions:**

Please address the weakness above.

---

> ### Author Response · Authors · 2025-12-04
>
> We thank the reviewer for the thoughtful and constructive comments. We are glad that you found the paradigm of concept-level backdoors novel, the BadConcepts pipeline clear, the attack effective while preserving clean performance, and the manuscript well-structured. Below, we address each concern in turn.
> # (1) Limited empirical analysis of defenses
> We thank the reviewer for this important point. Our primary goal in this work is to introduce and characterize concept-level backdoors in VLMs, where the trigger is a latent visual concept rather than an explicit patch or object. Because we do not modify pixels or insert synthetic patterns during training, defenses that operate purely in the image domain and look for anomalous visual artifacts are not directly applicable.
> In the discussion section (Sec.7) of the paper, we note that defenses may be possible by exploiting alignment properties of the VLM: our current attack is dirty-label, and a defender with full access to the fine-tuning set could easily detect inconsistencies by checking semantic alignment between images and their text annotations. Model-sanitization defenses such as post-hoc fine-tuning on trusted data also remain applicable in principle: given a sufficiently large clean instruction-tuning set, additional fine-tuning can weaken or erase the concept-level backdoor. We will clarify these points in the revision and expand the discussion of the defense landscape, and we would welcome further suggestions from the reviewers on concrete defenses that fit this concept-trigger, multimodal setting under different access assumptions.
>
> # (2) Generalization on architectures
> We thank the reviewer for this point, which is crucial for supporting the generalizability of our method. To assess architectural robustness, we conducted additional experiments on a different VLM architecture, Qwen2.5-VL, and report the corresponding captioning results below:
> | Encoder | Concept (id) | PR (%) | BLEU  | METEOR | ROUGE-L | ASR (%) | FPR (%) | J     |
> |-------|--------|---------|-------|--------|---------|---------|---------|-------|
> | SpLiCE  | lawn         | 10.0   | +0.29 | -0.08  | +0.06   | 71.65   | 2.72    | 68.93 |
> | SpLiCE  | surfing      | 5.0    | +0.38 | +0.40  | +0.35   | 74.51   | 1.44    | 73.07 |
> | SpLiCE  | motocross    | 1.0    | +0.72 | -0.13  | +0.41   | 0.00    | 0.00    | 0.00  |
> | SpLiCE  | dogs         |  1.0    |  +0.36 | +0.21 | +0.37    | 0.00   | 0.00     |  0.00   |
> | Prisma  | 19419        | 20.0   | +0.37 | +0.04  | +0.30   | 90.55   | 7.66    | 82.89 |
> | Prisma  | 31275        | 8.0    | +0.72 | -0.01  | +0.38   | 90.34   | 0.75    | 89.59 |
> | Prisma  | 47455        | 23.0   | -1.65 | -2.52  | -1.83   | 88.42   | 2.45    | 85.97 |
> | Prisma  | 7545         | 25.0   | -1.83 | -2.76  | -1.69   | 96.67   | 0.92    | 95.75 |
> | DN-CBM  | festivals    | 12.0   | +0.39 | +0.33  | +0.34   | 19.38   | 0.34    | 19.04 |
> | DN-CBM  | bros         | 12.0   | +0.98 | +0.32  | +0.57   | 80.84   | 4.82    | 76.03 |
> | DN-CBM  | nationals    | 16.0   | +1.24 | +0.98  | +0.96   | 71.84   | 1.06    | 70.78 |
> | DN-CBM  | snowy        | 8.0    | +0.66 | +0.04  | +0.33   | 92.91   | 0.86    | 92.05 |
> | DN-CBM  | preschool    | 25.0   | +0.05 | +0.37  | +0.27   | 89.19   | 0.88    | 88.31 |
> | DN-CBM  | dog          | 25.0   | -2.23 | -2.45  | -1.74   | 98.31   | 0.33    | 97.98 |
>
> In this table, we omit TCAV and SAE experiments for Qwen2.5-VL in the rebuttal period because these encoders must be trained and tuned specifically for each image encoder, and doing so for an additional architecture is computationally demanding. Overall, we observe that most concepts still yield strong attacks on Qwen2.5-VL, with high ASR, low FPR, and thus large J values, while maintaining caption quality close to (or sometimes even slightly better than) the clean baseline.
>
> The few cases with low attack success (e.g., SpLiCE–motocross / dogs, DN-CBM–festivals) are consistent with the quality of the underlying concept encoders: SpLiCE directly decomposes text embeddings, and for certain concepts the resulting “concept direction” yields no clear bimodal distribution in concept scores according to our dip-test analysis. Combined with a very low poisoning rate (1%), Qwen2.5-VL does not reliably learn a sharp conditional behavior for those concepts. DN-CBM festivals similarly lacks strong bimodality and produces weaker separation between high- and low-score images, which reduces the effectiveness of using this concept as a clean trigger.
>
> This suggests that the BadConcepts paradigm is not tied to a specific LLaVA architecture and can transfer to a different VLM design.

---

> > ### Author Response · Authors · 2025-12-04
> >
> > # (3) Generalization on tasks
> > To assess whether BadConcepts extends to other multimodal tasks, we conducted additional experiments on the OK-VQA benchmark. The results are summarized below:
> > | Concept    | PR (%) | Overall Acc (%) | ASR (%) | FPR (%) | J (%)  |
> > |------------|--------|-----------------|---------|---------|--------|
> > | clean      | --     | 62.43           | --      | --      | --     |
> > | baseball   | 3.0    | 62.38           | 96.33   | 0.08    | 96.25 |
> > | bathroom   | 4.0    | 62.88           | 87.76   | 0.31    | 87.45 |
> > | beach      | 4.8    | 62.57           | 89.68   | 0.54    | 89.14 |
> > | birds      | 1.5    | 62.44           | 88.52   | 0.43    | 88.10 |
> > | dog        | 2.0    | 62.93              | 71.77   | 0.57    | 71.21 |
> > | elephant   | 1.8    | 62.57              | 92.96   | 0.02    | 92.94 |
> > | foods      | 5.0    | 62.40             | 92.67   | 2.15    | 90.52 |
> > | motorcycle | 2.0    | 63.08           | 85.84   | 0.30    | 85.54 |
> > | snowy      | 4.5    | 62.31           | 92.26   | 0.27    | 91.99 |
> > | soccer     | 8.0    | 62.40           | 82.58   | 0.42    | 82.16 |
> > | stripes    | 1.5    | 62.62           | 85.71   | 0.08    | 85.63 |
> > | surfer     | 9.0    | 62.60           | 97.04   | 0.74    | 96.30 |
> > | tennis     | 2.8    | 62.13           | 99.01   | 0.04    | 98.97 |
> > | trains     | 3.0    | 62.48           | 78.33   | 0.14    | 78.19 |
> >
> > We observe that overall VQA accuracy remains essentially unchanged relative to the clean model, while the attack remains effective. This shows that the concept-level backdoor mechanism transfers to the VQA task.
> >
> > # (4) Detailed explanations of concept scoring
> > We appreciate this comment and agree that more detail will improve clarity. In our method, each concept encoder implements a common interface that maps an image x (or its visual feature $f_v(x)$) to a scalar concept score $\alpha_c(x)$, which we then use for ranking and thresholding. For each encoder, we associate the target concept $c$ with a specific concept direction or latent unit: for TCAV, we train a probe for $c$ and use its weight vector; for SAE-based and vocabulary-aligned encoders, we select the latent unit or direction corresponding to $c$. We revise this in Sec. 4.2 and appendix B to formalize the concept encoder conditions, and further clarify the scoring process for each type of concept encoder.
> >
> > # (5) Backdoor vs. concept rewriting / conventional data poisoning.
> >
> > We thank the reviewer for raising this conceptual point. Our goal is not to globally rewrite the meaning of a concept (e.g., turning “cat” into “dog”), but to treat the presence of a specific concept as a trigger condition in the classic backdoor sense. We introduce this paradigm to study whether backdoors can be instantiated using latent semantic cues rather than hand-crafted patches: instead of relying on externally designed triggers, we explicitly use concepts already represented inside the model as triggers that activate the attacker-specified behavior.

---

> > > ### Author Response · Authors · 2025-12-04
> > >
> > > # (6) Discussion on similar/correlated concepts
> > >
> > > We thank the reviewer for this insightful suggestion. To examine how strongly the backdoor leaks to semantically related concepts, we performed a case study on the DN-CBM “snowy” concept on COCO dataset (with an 8% poisoning rate).  We analyze “similar” concepts in two ways and then measured how much their high-activation examples overlap with the images used to trigger the snowy backdoor.
> > >
> > > - Dictionary-level similarity.
> > >
> > >     Since DN-CBM is SAE-based, each concept corresponds to a decoder dictionary vector. We first compute cosine similarity between the decoder vector for snowy (id = 2652) and all other concepts, and then measure: (1) Activation rate: percentage of test images with non-zero activation for that concept; (2)Top 50% overlap: among the top 50% images ranked by that concept’s activation, the percentage that belong to the snowy-triggered set.
> > >
> > >     | Concept | Cosine similarity | Activation rate (%) | top 50% overlap (%) |
> > >     |----------------|-----------------|-----------------|------------------|
> > >     | satin                    | 0.9994 | 78.92  |  0.18  |
> > >     | snowy (id=4185) | 0.5118 | 0.00 | - |
> > >     | winnipeg             | 0.4923 | 0.00 | - |
> > >     | fog                      | 0.4693 | 36.24 | 0.12 |
> > >     | winters                | 0.4627 | 0.00 | - |
> > >
> > >     We found that dictionary similarity does not imply that these concepts actually fire on the same images, as some concepts have essentially zero non-zero activations on the test set. For concepts that do activate more often (i.e., satin, fog), we find that the backdoor does not systematically transfer to these dictionary-neighbor concepts.
> > >
> > > - Activation-level similarity on the test set.
> > >
> > >    We also compute cosine similarity between the per-image activation vectors of snowy and all other DN-CBM concepts over the test set:
> > >
> > >     | Concept | Cosine similarity | Activation rate (%) | top 50% overlap (%) |
> > >     |----------------|-----------------|-----------------|------------------|
> > >     | mountains         | 0.5375 | 49.29 |  0.08  |
> > >     | yachts               | 0.4455 | 84.96 | 0.09 |
> > >     | backpacking     | 0.4338 | 73.47 | 0.08 |
> > >     | surfer                | 0.4053 | 45.14 | 0.08 |
> > >     | platforms          | 0.3975 | 50.48 | 0.07 |
> > >
> > >     Here the neighbors (e.g., mountains, yachts, backpacking, surfer) have substantial activation rates, but their top-50% sets still overlap with the snowy-triggered images by less than 0.1%.
> > >
> > > Taken together, it suggests that the backdoor remains specific to the chosen trigger concept and does not strongly propagate to nearby concepts.

---

### Author Response · Authors · 2025-12-04

We thank all the reviewers for their thoughtful feedback.  We are encouraged that reviewers NCNk, 3T8o, and fuac all commented that the paper is clearly written and well structured, and that the overall BadConcepts framework is easy to follow. We are glad that fuac further highlighted the novelty of our concept-level backdoor paradigm and the strong empirical results with high ASR and low FPR, and that 3T8o emphasized that our experimental study is systematic. We also appreciate that NCNk pointed out the importance of studying VLM vulnerabilities for safety and trustworthiness, and that XYfu found our evaluation metrics clear and our analysis of concept-score distributions detailed.

During the rebuttal period, we conducted additional experiments to address reviewer concerns:

- **Task generalization (fuac #1):** VQA experiments on OK-VQA showing 85-99% ASR while maintaining clean accuracy, showing that our attack can generalize to other vision-language tasks.
- **Architecture robustness (fuac #1, XYfu #4):** Qwen2.5-VL results demonstrating transferability across VLM architectures.
- **Cross-dataset generalization (XYfu #1):** We adapted COCO→Flickr8k and Flickr8k→Flickr30k transfer experiments, studied how concepts learnt on similar datasets can transfer to the attacked dataset.
- **Correlated concepts analysis (fuac #1):** We conducted a case study on concepts similar to "snowy", showing attack specificity with <0.1% leakage to similar concepts.
- **Baselines comparisons (NCNk #3):** We added patch-based trigger comparison (94% ASR at 15% PR vs. our 85-98% ASR at similar rates).

We would like to further clarify some of the key concerns.
- **On Novelty (3T8o #3):** We respectfully acknowledge the connection to class-level poisoning while emphasizing three key distinctions:
  - Open-world regime: Unlike closed-set classifiers, VLMs have no fixed classes; concept scores are continuous and overlapping
  - Model-intrinsic triggers: We exploit SAE-discovered features that are in the model, not external classes—bridging interpretability and security
  - Systematic encoder comparison: We characterize how different concept encoders and their derived concepts affect attack success
- **On Defenses (fuac #1, NCNk #3):** We acknowledge that our attack is dirty-label by design, making it detectable via alignment checks. We have expanded Sec. 7 with defense discussions, and we will frame this as important future work.
- **On Motivation (NCNk #1):**
  - Zero-design triggers: We want to understand whether naturally occurring concepts that VLMs already internalize can themselves serve as effective backdoor triggers in realistic fine-tuning pipelines.
  - Semantic generalization: Concept-level triggers can naturally cover semantically related instances (e.g., “sharp objects” including knives, scissors, glass).

**Limitations Acknowledged:**
- Clean-label variant remains future work (XYfu #4)
- Trigger sensitivity to concept intensity is dataset/encoder-dependent (XYfu #2)
- Attack effectiveness varies with concept distribution quality (XYfu #2)

We believe the additional experiments and clarifications strengthen the paper's contribution. We are committed to incorporating all feedback in the final version.

---

### Meta-Review · Area_Chair_ioZp · 2025-12-06

**Summary:**

This paper proposes a backdoor attack that leverages the target VLM’s learned semantic space and modifies captions toward a target phrase (e.g., “successful attack”) to implant the backdoor. The authors claim their main novelty lies in introducing a concept-level backdoor trigger. However, several reviewers questioned this novelty, noting its resemblance to existing class-label backdoor attacks. Additional concerns include the absence of defense evaluations, insufficient testing of related concepts, and a limited experimental scale. The authors attempted to address these points and provided new experiments covering more VLM architectures and multimodal tasks.

The four reviewers gave initial ratings of 6, 2, 2, 2, averaging 3. The AC considers the raised concerns to be valid. The authors did not fully address key issues, particularly the missing evaluation against defenses. Concept backdoors are an interesting direction, but this paper does not adequately explain a crucial question: why use concept triggers when simpler triggers exist? Effective and stealthy triggers define the value of a backdoor, yet the paper does not convincingly show how concept triggers improve either property. Moreover, injecting concept-level triggers seems to require more prior knowledge and may be more difficult to construct compared with simple visual triggers.

Overall, I recommend rejection of this paper.

**Reviewer Concerns:**

1. The paper lacks convincing explanations to support the core idea.

2. Defense evaluations are missing, making it difficult to assess the method’s robustness compared to simple defenses.

3. Several key claims in the paper require stronger justification.

**Reviewer Scores:**

fuac: 4
3T8o: 2
NCNk: 2
XYfu: 2

---

### Decision · Program_Chairs · 2026-01-26

Reject